# PCM1 is necessary for focal ciliary integrity and is a candidate for severe schizophrenia

Tanner O. Monroe [1,2], Melanie E. Garrett[3], Maria Kousi[4], Ramona M. Rodriguiz[5,6], Sungjin Moon[7], Yushi Bai[8], Steven C. Brodar[8], Karen L. Soldano[3], Jeremiah Savage[9], Thomas F. Hansen[10,11], Donna M. Muzny[12,13], Richard A. Gibbs[12,13], Lawrence Barak[8], Patrick F. Sullivan[14,15,16], Allison E. Ashley-Koch [3], Akira Sawa [17,18,19,20], William C. Wetsel [5,6,8,21], Thomas Werge [10,11,22,23] & Nicholas Katsanis [1,2✉]

The neuronal primary cilium and centriolar satellites have functions in neurogenesis, but little is known about their roles in the postnatal brain. We show that ablation of pericentriolar material 1 in the mouse leads to progressive ciliary, anatomical, psychomotor, and cognitive abnormalities. RNAseq reveals changes in amine- and G-protein coupled receptor pathways. The physiological relevance of this phenotype is supported by decreased available dopamine D2 receptor (D2R) levels and the failure of antipsychotic drugs to rescue adult behavioral defects. Immunoprecipitations show an association with Pcm1 and D2Rs. Finally, we sequence *PCM1* in two human cohorts with severe schizophrenia. Systematic modeling of all discovered rare alleles by zebrafish in vivo complementation reveals an enrichment for pathogenic alleles. Our data emphasize a role for the pericentriolar material in the postnatal brain, with progressive degenerative ciliary and behavioral phenotypes; and they support a contributory role for *PCM1* in some individuals diagnosed with schizophrenia.

[1] Department of Pediatrics, Northwestern University Feinberg School of Medicine, Chicago, IL 60611, USA. [2] Advanced Center for Translational and Genetic Medicine (ACT-GeM), Stanley Manne Children's Research Institute, Ann & Robert H. Lurie Children's Hospital of Chicago, Chicago, IL 60611, USA. [3] Duke Molecular Physiology Institute, Duke University School of Medicine, Durham, NC 27701, USA. [4] MIT Computer Science and Artificial Intelligence Laboratory (CSAIL), Broad Institute of MIT and Harvard, Cambridge, MA 02139, USA. [5] Department of Psychiatry and Behavioral Sciences, Duke University School of Medicine, Durham, NC 27710, USA. [6] Mouse Behavioral and Neuroendocrine Analysis Core Facility, Duke University School of Medicine, Durham, NC 27710, USA. [7] Department of Biological Sciences, Kangwon National University, Chuncheon 24341, Republic of Korea. [8] Department of Cell Biology, Duke University School of Medicine, Durham, NC 27710, USA. [9] Center for Translational Data Science, The University of Chicago, Chicago, IL 60615, USA. [10] Department of Clinical Sciences, University of Copenhagen, Copenhagen, Denmark. [11] Institute of Biological Psychiatry, MHC Sct. Hans, Mental Health Services, Copenhagen, Denmark. [12] Human Genome Sequencing Center, Baylor College of Medicine, Houston 77030 TX, USA. [13] Department of Molecular and Human Genetics, Baylor College of Medicine, Houston, TX 77030, USA. [14] Department of Genetics, University of North Carolina at Chapel Hill, Chapel Hill, NC 27599, USA. [15] Department of Psychiatry, University of North Carolina at Chapel Hill, Chapel Hill, NC 27599, USA. [16] Department of Medical Epidemiology and Biostatistics, Karolinska Institutet, Stockholm SE-171 77, Sweden. [17] Department of Psychiatry, Johns Hopkins University School of Medicine, Baltimore, MD 21287, USA. [18] Department of Neuroscience, Johns Hopkins University School of Medicine, Baltimore, MD 21287, USA. [19] Department of Biomedical Engineering, Johns Hopkins University School of Medicine, Baltimore, MD 21287, USA. [20] Department of Mental Health, Johns Hopkins University School of Medicine, Baltimore, MD 21287, USA. [21] Department of Neurobiology, Duke University School of Medicine, Durham, NC 27710, USA. [22] iPSYCH - The Lundbeck Foundation Initiative for Integrative Psychiatric Research, Copenhagen, Denmark. [23] Center for GeoGenetics, GLOBE Institute, University of Copenhagen, Copenhagen, Denmark. ✉email: nkatsanis@luriechildrens.org

Schizophrenia (SZ) is a disorder with a complex genetic architecture[1]. Studies focusing on common allele associations or rare allele burden under a variety of experimental designs have made progress in discovering associated loci. However, these have required large sample sizes to reveal significant signals[2,3], indicating that SZ is genetically and clinically heterogeneous with multiple driving processes.

Among candidate biological pathways to emerge from genome-wide association and sequencing studies has been modulators of synaptic transmission; however, these account for a modest fraction of the genetic burden of SZ[2]. Centrosomal and ciliary dysfunction have also been highlighted among candidate pathways. For instance, common alleles near the known ciliopathy-encoding locus SDCCAG8 have been reported in case-control association studies and have reached genome-wide significance[2,4]. Locus-specific analyses have shown association with ciliary and centriolar loci including AHI1[5], NUDEL[6], PCNT[7], and PCM1[8]. This observation is unsurprising, since ablation of ciliary and centriolar proteins in the mouse causes significant neurogenesis and behavioral phenotypes. Moreover, Bardet-Biedl individuals (a model ciliopathy) with deleterious ciliary alleles exhibit a higher incidence of cognitive deficits and psychosis compared to the general population[9].

These findings have led to the hypothesis that the cilium—an organelle critical for paracrine signaling, microtubule organization, cell cycle regulation, and cell polarization—may also be relevant to the development of psychiatric disorders[10]. Several groups have reported trend-level associations of variants in pericentriolar material 1 (PCM1, OMIM (600299)) with SZ[8,11–13], with one study suggesting an association with a specific subset of SZ individuals with volumetric changes in the orbitofrontal cortex gray matter[8]. Although caution is warranted when interpreting gene-specific association tests, we noted that several known biochemical interactors with PCM1 include the credible SZ candidates SDCCAG8[14], RPGRIP1L[15] and PCNT[16]. In addition, we reported a nonsense PCM1 mutation in a SZ individual treated with clozapine, suggesting deleterious variants in ciliary genes may be enriched in cases affected by treatment-refractory SZ.

Here, we present murine, zebrafish, and patient data which highlight a role for PCM1 in a subset of SZ cases. We find that ciliary integrity progressively declines in adult $Pcm1^{-/-}$ mice, accompanied by adult-onset anatomical, behavioral, and psychomotor deficiencies. Those deficits are resistant to antipsychotic drugs. Next, we sequence two cohorts of individuals with severe SZ and use zebrafish in vivo complementation to discover an over-representation of rare pathogenic PCM1 variants. Our results suggest PCM1 plays a role in some SZ cases, and point to a postnatal role for the cilia in maintaining brain structure and function.

## Results

**$Pcm1^{-/-}$ mice exhibit adult anatomical and behavioral defects.** To examine the in vivo role of Pcm1, we developed a mutant mouse from embryonic stem cells harboring a gene-trap insertion (Supplementary Fig. 1a). Loss of Pcm1 was verified through mRNA, western blot, and by immunofluorescence (Supplementary Fig. 1b–e and Supplementary Fig. 7c). The offspring genotypes were consistent with Mendelian segregation, suggesting an absence of prenatal lethality. Given the established roles for ciliary and PCM-associated proteins in neurodevelopment and the fact that Pcm1 is expressed in the mouse ventricular zone during neurogenesis[17], we were surprised to discover that mice lacking Pcm1 failed to show overt congenital malformations. Although axonemal mouse mutants are often lethal prenatally, $Pcm1^{-/-}$ mice had no appreciable prenatal pathology and showed minimal

postnatal anatomical or functional sensory deficits (Supplementary Fig. 1f, g). Brain volumetric analyses at P0 (post-natal day 0) revealed the lateral ventricles and hippocampi to be indistinguishable from control littermates. Likewise, staining of cortical slices at P3 with antibody markers for layers II-IV (CUX1) and V (CTIP2) failed to reveal any gross lamination defects (Supplementary Fig. 1h).

Because of the possible association of PCM1 in SZ individuals with brain volumetric changes later in life[8], we repeated these analyses after P90. Brain sections from adult mice showed a clear increase in lateral ventricular size, which we quantified by stereology (Fig. 1a–c). Although this phenotype is not unique to SZ, it is one of the most consistent brain abnormalities in SZ[18]. Motivated by these sectioned tissue differences, we decided to quantify the phenotype in situ using magnetic resonance imaging (MRI) (Fig. 1d). We replicated the lateral ventricle size differences increase in the mutants compared to wild-type (WT) (Fig. 1e, f). In addition, corresponding to observations from individuals with SZ[19], we saw a reduced overall brain mass in adult $Pcm1^{-/-}$ mice (Fig. 1g, h). Our MRI studies revealed this decrease was predominantly driven by reductions in cortical volume (Fig. 1i, j), another consistent hallmark of SZ[20]. Although we could not detect a significant reduction in cortical layer thickness (Supplementary Fig. 2a, b), we found a lower neuronal density in $Pcm1^{-/-}$ cortex (Fig. 1k), which has been linked to disease risk[21]. Similarly, we detected a reduced striatal volume in $Pcm1^{-/-}$ mice (Fig. 1l, m). While there are differences in striatal neuroanatomy between mice and humans, striatal dysfunction is thought to be a fundamental underlying feature of SZ[22]. We suspected reduced hippocampal volume, based on the two-dimensional view of the tissue (Fig. 1a, Supplementary Fig. 2c), so we were surprised to find that 3-D MRI showed hippocampal volumes similar to WT. Histology revealed that the hippocampal subfields had similar thicknesses to the control (Supplementary Fig. 2d–f). However, we noticed an altered hippocampal morphology in the $Pcm1^{-/-}$ mice, which we quantified as the ratio of the length of the septal pole to the temporal pole. We found that $Pcm1^{-/-}$ mice had a relatively shorter septal pole (Supplementary Fig. 2e, g). This result is in contrast to other ciliopathies, which present with reduced hippocampal volumes[23,24].

Due to morphological abnormalities in $Pcm1^{-/-}$ mice similar to some SZ individuals, we conducted behavioral tests on adult P90-P120 mice. We quantified locomotion in the open field, as enhanced activity in rodents may reflect the hyperdopaminergia or hypoglutamatergia associated with SZ. Spontaneous activity was augmented in $Pcm1^{-/-}$ mice compared to WT littermates (Fig. 2a, b). To examine whether locomotor responses to psychostimulants was abnormal, mice were habituated for 60 min to the open field, administered acutely vehicle or different doses of phencyclidine (PCP), and returned immediately to the open field. Following injection PCP-stimulated hyperlocomotion in both genotypes was dose-dependent (Fig. 2c). However, $Pcm1^{-/-}$ mice were hyper-responsive to both doses of PCP relative to WT.

We quantified WT and $Pcm1^{-/-}$ littermates' prepulse inhibition (PPI), a test for sensorimotor gating which is abnormal in SZ as well as in several other psychiatric disorders. Although PPI increased in an intensity-dependent fashion in both genotypes, $Pcm1^{-/-}$ mice had reduced PPI at 8 and 12 dB prepulse intensities relative to WT (Fig. 2d). Null activities were similar between genotypes (Supplementary Fig. 3a), while startle responses were enhanced in mutant mice (Supplementary Fig. 3b). The hyper-activity in open field and the reduced PPI in adult $Pcm1^{-/-}$ mice suggest that mutants present with positive-like symptoms of SZ-like behaviors. In addition, responses to PCP in the open field

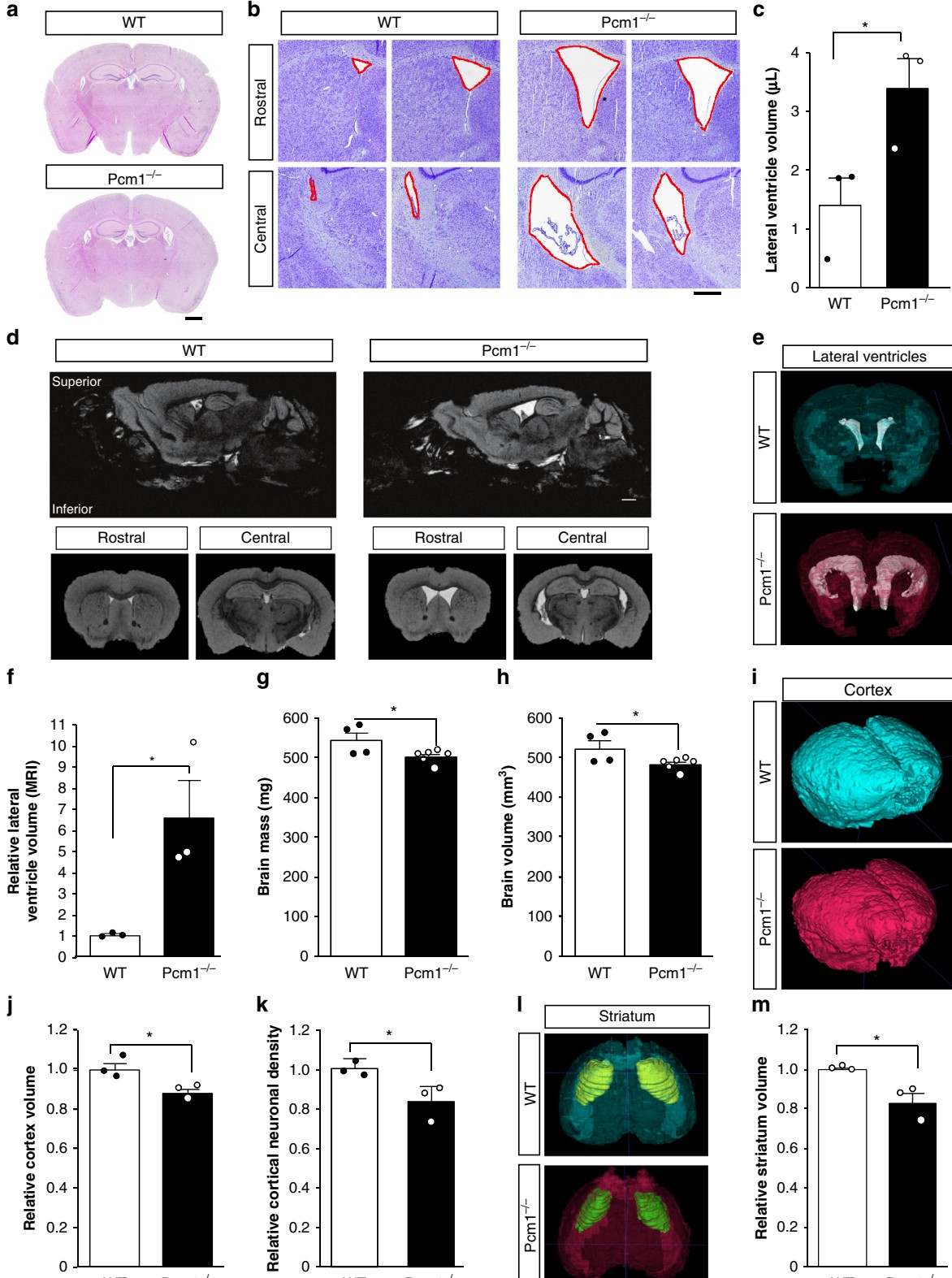

**Fig. 1 Neuroanatomical defects in adult *Pcm1*<sup>−/−</sup> mice. a** Representative H&E sections from control and *Pcm1*<sup>−/−</sup> brains. Scale 1 mm (**b**) Representative Cresyl violet histological sections from two WT and two *Pcm1*<sup>−/−</sup> brains depicting enlarged ventricles in the mutant. **c** Lateral ventricle volume by stereology ($N = 3$; $p = 0.047$). Scale 0.4 mm (**d**) Representative slices from adult WT and *Pcm1*<sup>−/−</sup> brain MRIs. Scale 0.6 mm (**e**) Representative volumetric views of the lateral ventricles. **f** Lateral ventricle volume ($N = 3$; $p = 0.036$). **g** Total brain size ($N = 4$ and 6); $p = 0.05$). **h** Total brain volume ($N = 4$ and 6; $p = 0.05$). **i** Representative volumetric views of the cortex. **j** Cortex volume ($N = 3$; $p = 0.035$). **k** Cortical neuron density ($N = 4$; $p = 0.045$). **l** Representative volumetric views of the striatum (anterior view, shown with transparent cortex). **m** Striatal volume ($N = 3$; $p = 0.027$). Data presented as mean ± SEM; *$p < 0.05$, WT vs. *Pcm1*<sup>−/−</sup>. Two-sided *t*-tests, without adjustments. Source data and detailed statistical information are provided as a Source Data File.

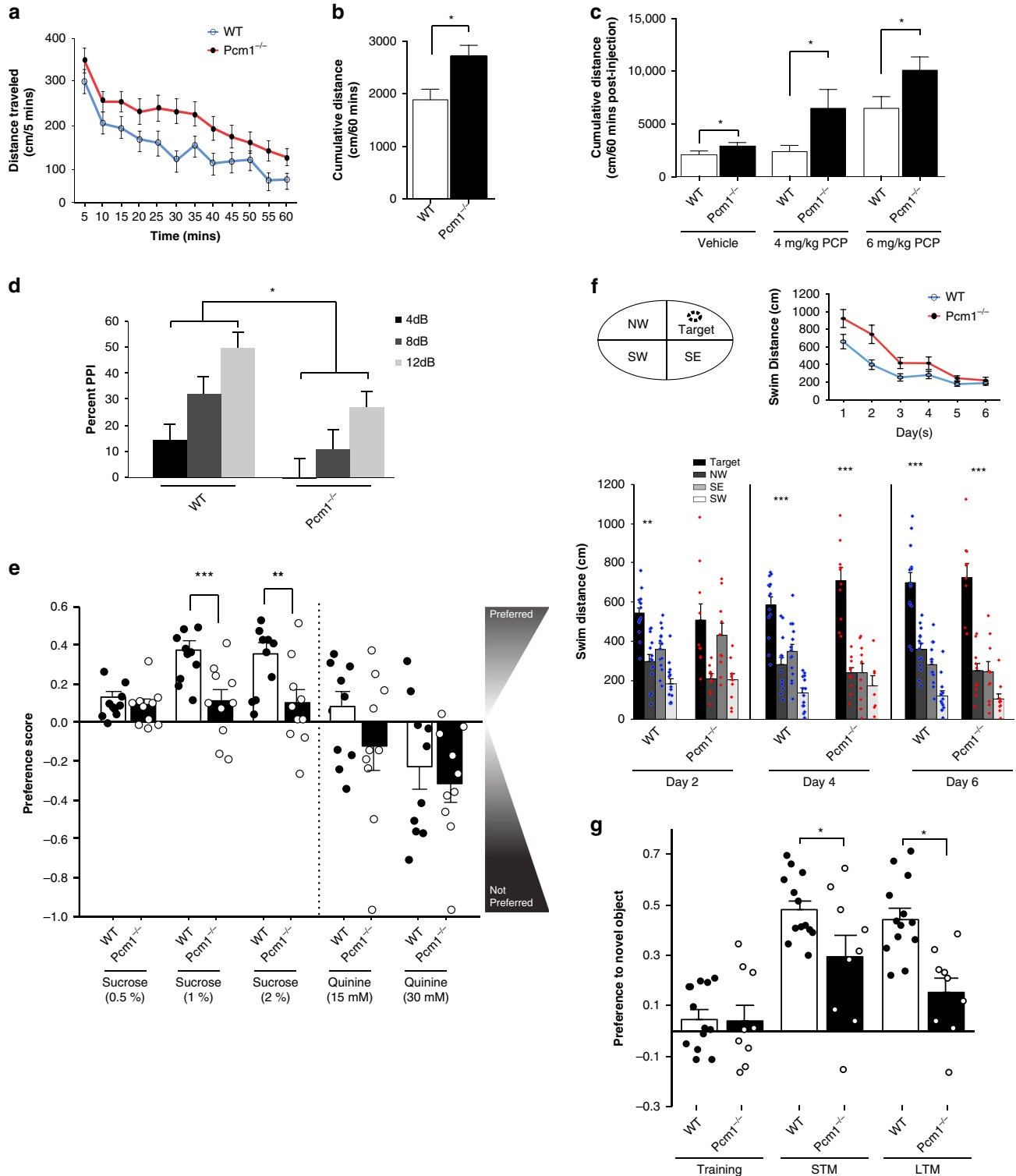

potentially suggest that dopaminergic and/or glutamatergic neurotransmission may be abnormal in $Pcm1^{-/-}$.

To assay for negative symptoms of SZ-like behaviors, we tested mice for anhedonia-like responses. WT and $Pcm1^{-/-}$ littermates housed individually were presented with two identical bottles containing water. Following stable intake, one of the bottles was replaced with a sucrose solution and we quantified their preferences. Across increasing concentrations of sucrose, WT mice showed the increased sucrose preference (Fig. 2e). However, the mutants displayed a reduced preference for 1% and 2%

sucrose relative to the WT. By comparison, responses to quinine (bitter taste) were not different between the genotypes, indicating that anhedonia-like behavior in the $Pcm1^{-/-}$ mice was not due to gustation abnormalities.

To quantify cognitive function, we subjected $Pcm1^{-/-}$ mice to the Morris water maze and to the novel object recognition memory tests (NORM). In the water maze, all mice learned the location of the hidden platform. However, during acquisition $Pcm1^{-/-}$ mice swam longer distances on days 2 and 3 than WT (Fig. 2f). On probe trials, we observed WT mice swam farther in

**Fig. 2 Behavioral defects in adult *Pcm1*$^{-/-}$ mice. a** Spontaneous activity in the open field at 5 min intervals over 60 min. **b** Cumulative distance travelled in the open field. (**a–b**, $N = 21$ WT, $N = 30$ *Pcm1*$^{-/-}$; $p < 0.001$ (time), $p = 0.04$ (genotype)). **c** Locomotor responses to PCP in the open field as cumulative distance travelled ($N = 11$ and 18 mice/genotype/treatment condition; $p = 0.04$). **d** Prepulse inhibition to the 4, 8, and 12 decibel (dB) prepulses ($N = 11$ and 12 mice/genotype; $p = 0.006$ (genotype), $p < 0.001$(PPI), $p = 0.013$ (genotype)). **e** Anhedonia-like responses in *Pcm1*$^{-/-}$ mice as determined in a two-bottle sucrose preference test with different concentrations of sucrose. Because responses to quinine were not distinguished by genotype, gustatory responses appeared similar in the two groups ($N = 10$ mice/genotype/repeated test condition; $p = 0.002$ (1% sucrose), $p = 0.013$ (2% sucrose)); overall effect of quinine concentration with all mice consuming less of the 30 mM quinine compared to water alone ($p = 0.015$); overall drinking volumes did not differ between genotypes and test conditions (range 8–10 ml/night/mouse). **f** Morris water maze performance showing the location of the hidden platform (Target) in the northeast (NE quadrant), as well as acquisition learning and probe trial performance. For acquisition an overall genotype effect for swim distance was found, $p = 0.035$. Probe trials showing swim distance in each quadrant, NE (Target), NW = northwest, SE = southeast, and SW = southwest quadrants. For probe tests significance denoted for target relative to remaining quadrants ($N = 10$ and 14 mice/genotype; within subjects contrast Probe Day × Genotype × Quadrant (quadratic), $p = 0.004$. No genotype differences detected. **g** Novel object recognition memory performance showing preference for the novel or familiar objects at training and on tests for short-term (STM) and long-term memory (LTM) ($N = 9$ and 13 mice/ genotype; RMANOVA Test Day × Genotype $p = 0.022$). Data presented as means ± SEM; behavioral data analyzed by *t*-tests, ANOVA, or repeated measures ANOVA with Bonferroni corrected *post-hoc* tests. *$p < 0.05$; **$p < 0.01$, ***$p < 0.001$, WT vs. *Pcm1*$^{-/-}$. Source data and detailed statistical information are provided as a Source Data File.

the target than in the other three quadrants on days 2, 4, and 6. By contrast, on day 2 *Pcm1*$^{-/-}$ mice failed to distinguish the target from the southeast quadrant. By days 4 and 6 these mutants learned the location of the hidden platform. In the NORM, at training both WT and *Pcm1*$^{-/-}$ mice did not prefer one of the identical objects over the other (Fig. 2g). However, when one of the familiar objects was replaced with a novel object 30 min or 24 h after training, both short- (STM) and long-term memory (LTM) were observed to be deficient in the *Pcm1*$^{-/-}$ mice. As control, we examined the total numbers of object contacts and the duration of contacts across training, and STM and LTM testing (Supplementary Fig. 4a, b). Since no significant genotype differences were observed at any of these times, the deficiency in episodic STM and LTM cannot be attributed to failure to interact with the objects. These findings indicate that *Pcm1*$^{-/-}$ mice are deficient in spatial and episodic memory and they further suggest a role for centriolar satellites in phenotypes related to SZ.

**Spatiotemporal degenerative ciliary defects in *Pcm1* mutants.** Previous studies showed that *PCM1* is necessary for ciliogenesis in vitro and suppression of its mRNA can lead to loss of cilia and/ or a reduction in cilia length[25,26]. Complete loss or severe shortening of cilia is incompatible with mammalian life[10]; however, *Pcm1*$^{-/-}$ animals survive gestation, suggesting that the initiation of cilia growth is not regulated solely by *Pcm1*. We assayed for ciliary defects in the CNS at four stages of post-natal development: P4 neonates, P21 juveniles, P40 prepubescent mice, and P90 adults. Using an antibody against type 3 adenylyl cyclase (AC3) that marks the length of the ciliary axoneme in neurons, we observed normal-appearing cilia at P4 (Supplementary Fig. 5a). By P21 we observed bulbous cilia in the amygdala and prelimbic cortex (Fig. 3a, d). By P40, cilia were shorter in the amygdala and prelimbic cortex, as well as in the CA1 hippocampus (but not in the CA3 and dentate gyrus) (Fig. 3b, e). At P90 the distribution of aberrant cilia remained localized spatially to the sites observed at P40; however, the frequency increased and included the septal nuclei and striatum (Fig. 3f, Supplementary Fig. 5b, c; Supplementary Fig. 10). Counter-staining AC3-positive cilia with a second ciliary marker (Arl13b) showed a similar pattern (Supplementary Fig. 5c), indicating that cilia, and not localization of AC3 protein, were defective. This spatial and temporal appearance of irregular cilia was unanticipated as *PCM1/Pcm1* is expressed ubiquitously in the CNS during human brain development (Human Brain Transcriptome[27]) and in the mouse brain (Allen Brain Atlas[28]) (Supplementary Fig. 5d, e). The apparently progressive nature of the structural ciliary defects

in our mutants raises the question as to whether adult behavioral phenotypes are also progressive. Correspondingly, we found no behavioral differences in spontaneous locomotor activity in open field, PPI, or NORM in juvenile (P21) mice (Supplementary Fig. 5a–c, e). Similarly, at pubescence (P40), we did not detect any genotype differences in spontaneous locomotor activity or anhedonia (Supplementary Fig. 5b, d). Nevertheless, a deficit in PPI emerged at P40, with responses in *Pcm1*$^{-/-}$ mice being significantly reduced at each prepulse intensity relative to WT (Supplementary Fig. 6c). Only null activities were reduced in p21 mutants whereas no other genotype effects for null or startle activities were found in the P21 and P40 mice (Supplementary Fig. 3c–f). Despite this fact, episodic STM and LTM were impaired at P40 in *Pcm1*$^{-/-}$ mice (Supplementary Fig. 4c–f; Supplementary Fig. 6e). The duration and number of contacts with objects was significantly higher in the P21 mutants while no genotype differences were observed in the P40 mice (Supplementary Fig. 5a–d). Collectively, these findings suggest that the ciliary defects in *Pcm1*$^{-/-}$ mice are progressive and may need to reach a critical level before behavioral phenotypes emerge.

**RNAseq and functional analyses reveal GPCR pathway defects.** To identify the subcellular abnormalities associated with the gross morphological and behavioral defects of our mutants, we performed RNAseq on frontal cortex, striatum, and hippocampus from *Pcm1*$^{-/-}$ and WT littermates. Hierarchical clustering revealed the highest correlation between samples of the same anatomical region, with pairwise differential expression between genotypes showing clear reproducibility within those regions (see Methods, Fig. 4a, Supplementary Fig. 7a, b). As expected, *Pcm1* was the most significantly down-regulated gene across all regions (Supplementary Fig. 7c). We then performed Gene Set Enrichment Analysis (GSEA[29]) to discover gene ontology (GO) pathways significantly different between genotypes. Across the tissues, we identified significant changes in neurological signaling pathways (Supplementary Table 2). Strikingly, across all three regions, we found two pathways consistently mis-regulated: G-protein coupled receptor (GPCR) activity and amine receptor activity (Fig. 4b). We also evaluated a specific cohort of genes reported to be transcriptionally altered in individuals with SZ[30] and found a significant overlap between this set and the genes in *Pcm1*$^{-/-}$ frontal cortex and hippocampus (Supplementary Table 3). One function of the primary cilium is to expose various receptors and other sensors to the extracellular environment[10]. In the context of the neuronal cilium, a prominent role for sensing environmental stimuli has been ascribed to multiple GPCRs. For example, the melanin-concentrating hormone receptor 1 (MCHR1) and

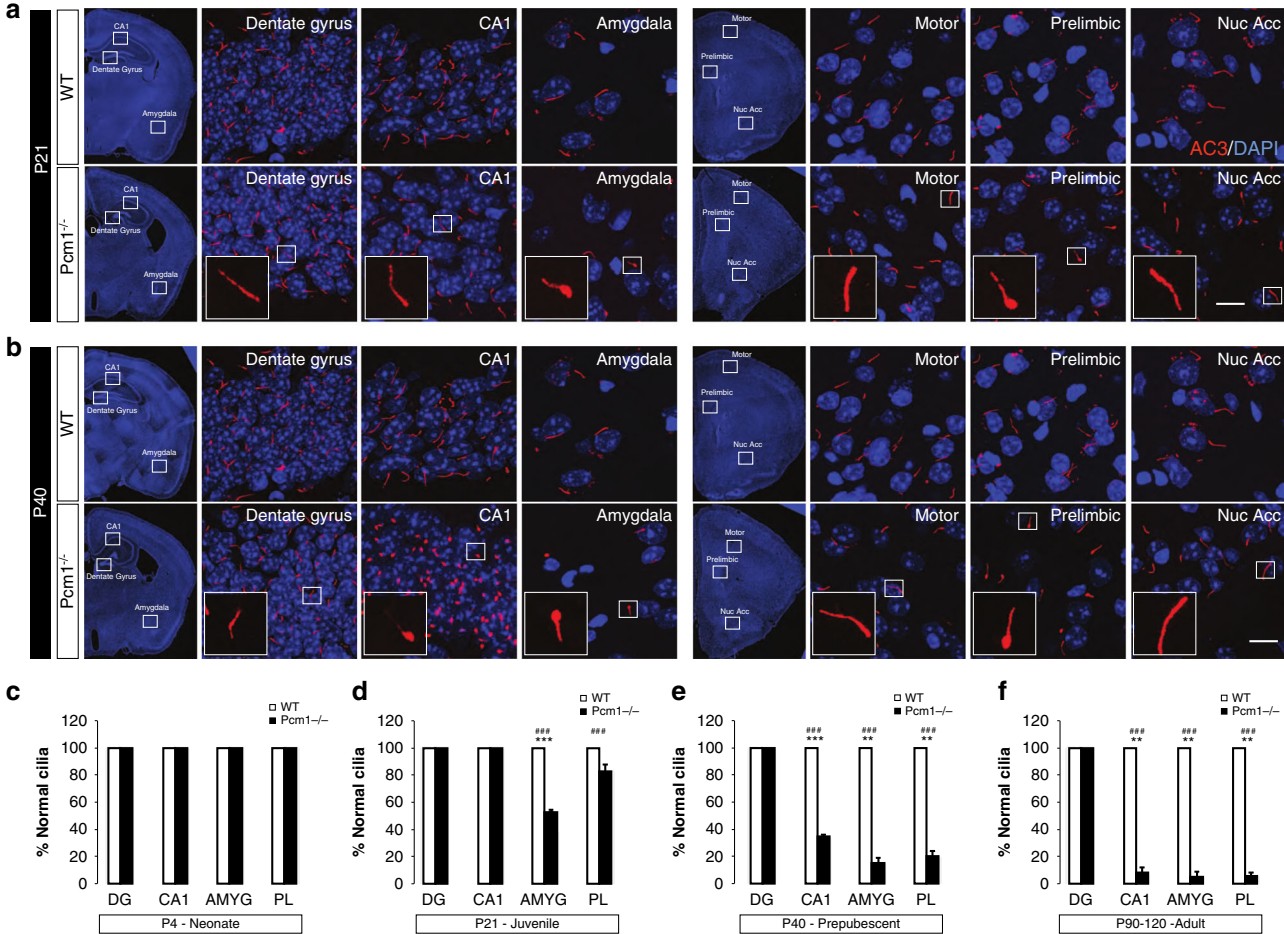

**Fig. 3 Temporal and spatial ciliary abnormalities at different ages. a** Coronal brain images revealing bulbous cilia in the prelimbic cortex and amygdala of P21 $Pcm1^{-/-}$ mice. Scale 5 μm (**b**) At P40, brain regions in the $Pcm1^{-/-}$ mice showing an increased frequency of bulbous cilia. Scale 5 μm (**c–f**) Bar graphs showing the proportion of normal cilia in various brain regions from P4 (**c**), P21 (**d**), P40 (**e**), and P90 (**f**) in WT and $Pcm1^{-/-}$ mice (**c–f**; $N > 30$ cilia/region/mouse, 3 mice/genotype; $t$-test $p < 0.001$ P21 AMYG; $p < 0.001$ P21 CA1; $p = 0.004$ P40 AMYG; $p = 0.005$ P40 PL; $p = 0.004$ P90 CA1; $p = 0.003$ P90 AMYG; $p = 0.0012$ P90 PL). DG dentate gyrus, AMY amygdala, PL prelimbic cortex. $\chi^2$ test, ###$p < 0.001$; $t$-tests, mean ± SEM **$p < 0.01$, ***$p < 0.001$, WT vs. $Pcm1^{-/-}$. Source data and detailed statistical information are provided as a Source Data File.

somatostatin type 3 receptor (SSTR3) localize to the axoneme of neuronal cilia and are mis-localized in ciliopathy mouse models. We reasoned that transcriptional changes in GPCRs in $Pcm1^{-/-}$ brain might lead to protein trafficking defects and, thereby, contribute to the anatomical and behavioral phenotypes observed in these mutants. However, testing with a variety of reagents failed to reveal a ubiquitous defect. Similar to the AC3/Arl13b staining shown previously (Supplementary Fig. 5c), SSTR3 and MCHR1 staining showed overlapping fluorescence with AC3 in cilia from various regions in the adult $Pcm1^{-/-}$ brain (Supplementary Fig. 8). Although it is possible that either GPCR might exhibit a spatial deficit in the axoneme, such alterations are probably secondary to a structural ciliary defect—especially since we did not observe inappropriate distribution of either SSTR3 or MCHR1 in the cytoplasm, a hallmark of ciliary trafficking defects[31].

We were struck by the uniform identification of amine pathway defects in our RNAseq data. This was particularly notable because of three observations. First, behavior defects in our $Pcm1^{-/-}$ mice can be explained partially by defects in dopamine signaling. Second, some dopamine receptors are known to be localized to primary cilia in the brain[32]. Finally, prior studies have shown other components of the pericentriolar material (e.g., DISC1) complex with the dopamine D2 receptor

(D2R)[33]. From these perspectives, we asked whether PCM1 resides in complex with D2R and/or the dopamine D1 receptor (D1R) by expressing HA-tagged PCM1 and Flag-tagged D2R and D1R in a hippocampal cell line (HT22). Immunoprecipitation (IP) of Flag-tagged D2R and D1R revealed that only D2R was complexed with PCM1 (Fig. 4c). To determine whether this association existed in vivo, we performed IPs for the D1R and D2R, and probed for Pcm1. Again, we observed a Pcm1 immunoreactive band only in the presence of D2R, suggesting these two proteins form a complex in vivo (Fig. 4c). Given the altered transcriptional levels of the D2R and the loss of $Pcm1$, we hypothesized that protein levels of D2Rs may be altered upon disruption of Pcm1. To evaluate this point, we quantified striatal D2Rs using saturation binding with the selective radiolabelled antagonist, [³H]-raclopride. Consistent with our behavioral and RNAseq analyses, our results revealed an ~25% decrease both in D2R Bmax binding and Kd values, suggesting that both protein levels and binding affinity of the receptors were altered in the $Pcm1^{-/-}$ mice (Fig. 4d).

**Impaired responses of $Pcm1^{-/-}$ mice to antipsychotics.** The $Pcm1^{-/-}$ mice have a series of related characteristics. For instance, $Pcm1^{-/-}$ striatal membranes have reduced D2R binding

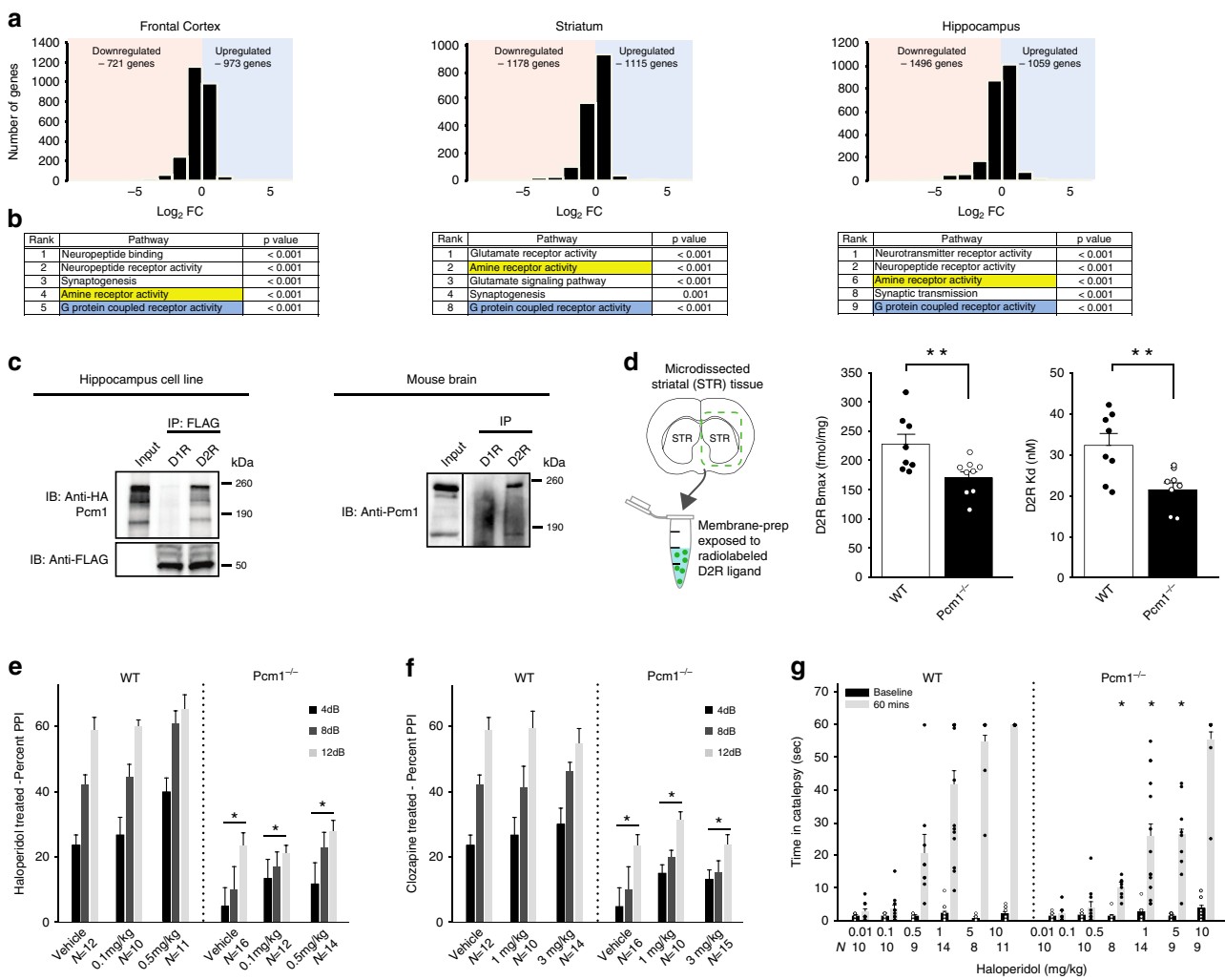

**Fig. 4 Altered dopaminergic signaling components in *Pcm1*−/− mice. a** mRNA sequencing from three brain regions in WT and *Pcm1*−/− adults ($N = 3$/genotype, 3 regions per subject): total number of genes up-regulated and down-regulated are plotted on a log₂FC scale. **b** Gene set enrichment analysis revealed amine and G protein coupled receptor activity pathways were altered across the three brain regions. **c** Immunoprecipitations for D1R (dopamine D1 receptor) and D2R (dopamine D2 receptor), followed by western blot for Pcm1 in mouse hippocampal cell lines and mouse brain homogenates. **d** Striatal (STR) membrane preparations from WT and mutant brains were dissected, membranes were isolated, and binding studies were instituted with the radiolabeled D2R antagonist [³H]-raclopride. Both the D2R Bmax and Kd levels were reduced significantly in samples from mutant mice ($N = 8$ WT, $N = 9$ *Pcm1*−/−; $p = 0.01$ (Bmax); $p = 0.004$ (Kd)). ANOVA with post-hoc Bonferroni adjustment. **e, f** PPI responses from P90-120 WT and *Pcm1*−/− mice indicate that mutants fail to respond at WT levels upon exposure to different doses of haloperidol (**e**) (Overall genotype effect, $p = 0.001$ PPI × genotype) or clozapine (**f**) (Overall genotype effect, $p = 0.001$). **g** Catalepsy in WT and *Pcm1*−/− mice after 0.5, 1, 5, and 10 mg kg⁻¹ haloperidol ((RMANOVA Pre/Post × Treatment × Genotype, $p = 0.002$)). **e–g** N shown in graph, analyzed as RMANOVA with post-hoc Bonferroni adjustment. Data shown as mean ± SEM, *$p < 0.05$, WT vs. *Pcm1*−/−. Source data and detailed statistical information are provided as a Source Data File. RNAseq data has been made publicly available. Illustrations are original.

levels and affinity relative to WT membranes; frontal cortex, striatum, and hippocampus from *Pcm1*−/− mice show changes in transcriptional expression of dopaminergic receptors; and these mutants are hyperactive in the open field and deficient in PPI. Collectively, these results led us to ask whether acute administration of antipsychotic drugs could ameliorate behavioral deficit in adult (P90) *Pcm1*−/− mice. We tested PPI responses in WT and *Pcm1*−/− mice administered different doses of haloperidol. Haloperidol treatment was not different from the vehicle control in the *Pcm1*−/− mice, leaving the mutants still impaired relative to the WT (Fig. 4e). Null activity was significantly increased in *Pcm1*−/− mice given 0.1 mg kg⁻¹ haloperidol, whereas startle activities were enhanced overall in mutants across the different doses of the neuroleptic (Supplementary Fig. 9a, b). These findings show that the *Pcm1*−/− mice are resistant to the actions of

the typical antipsychotic drug, haloperidol. Importantly, altered dopaminergic signaling in SZ has been attributed to increased dopamine synthesis in treatment-responsive SZ cases[34], as antipsychotics are used to block D2Rs. Therefore, it follows that haloperidol has little effect in *Pcm1*−/− mice, which already have reduced membranous D2R.

To determine whether *Pcm1*−/− mice were resistant also to effects of an atypical antipsychotic drug, we analyzed their responses to clozapine. Our results showed again that mutants were impaired in PPI and neither the 1 nor 3 mg kg⁻¹ doses could normalize their responses (Fig. 4f). As in all other PPI experiments, startle activities were significantly higher in the *Pcm1*−/− than in WT mice (Supplementary Fig. 9a, b). Collectively, these results indicate that in the *Pcm1*−/− mice PPI is resistant to effects of both haloperidol and clozapine.

To examine this relationship in more detail, we asked whether loss of Pcm1 would render the mutants responsive to haloperidol in catalepsy since this assay examines D2R post-synaptic responses. Both WT and $Pcm1^{-/-}$ mice were tested for catalepsy at time 0 (baseline). Subsequently, they were injected with different doses of haloperidol and tested 60 min after the injection (Fig. 4g). Catalepsy at baseline was low and not distinguished by genotype or by the dose to be assigned. However, 60 min after injection catalepsy was evident in both genotypes. In WT mice, haloperidol produced a dose-dependent increase in catalepsy from 0.5 to 10 mg kg$^{-1}$, with the 10 mg kg$^{-1}$ dose increasing this response in all mice beyond the 60 sec cut-off. Although catalepsy was enhanced in $Pcm1^{-/-}$ mice with haloperidol, responses were significantly lower than those in WT with the 0.5, 1, and 5 mg kg$^{-1}$ doses (Fig. 4g). Collectively, these findings indicate that $Pcm1^{-/-}$ mice are resistant to the effects of haloperidol and clozapine, and catalepsy responses are less robust in the mutants than in WT.

**A contribution of rare PCM1 alleles in individuals with SZ.** Taken together, our phenotypic and behavioral data suggest PCM1 as a rational functional candidate for SZ. Despite previous supportive association studies[13], the largest SZ genome-wide common variant association studies published have not found evidence of an association for PCM1[2,35]. However, a recent analysis that included chromatin interaction maps with GWAS identified PCM1 as a risk gene for SZ, and found cilia in gene ontologies of SZ risk genes[36]. Moreover, one prior study claimed a stronger association in SZ individuals with gray matter volumetric deficits[8], a consistent phenotype of the $Pcm1^{-/-}$ mice. In addition, our pharmacological studies with mice suggested that PCM1 SZ variants might be enriched in individuals refractory to standard treatment with antipsychotics, potentially representing a subset of SZ cases.

In light of the pharmacological data in mice, we asked whether there exists a genetic contribution of PCM1 in SZ individuals with minimal response to antipsychotics. We sequenced all 39 exons of PCM1 in a Danish sample composed of 380 cases and 380 ancestry-matched controls[37] who had received trials (i.e., sufficient dose and duration) of greater than or equal to three different antipsychotic drugs, but had minimal clinical response and who had been committed to a psychiatric ward (viz., atypical SZ or ASZ). The case cohort was confirmed to match controls by Principal Component Analysis (PCA; Supplementary Fig. 10. To determine whether variants in PCM1 were associated with ASZ, we used gene-based optimal sequence kernel association test (SKAT-O)[38]. Inclusion of 293 variants with a variant call rate >15%, regardless of minor allele frequency (MAF), provided no significant association for PCM1 variants with ASZ ($p = 0.068$; Table 1). Because variants with strong functional effect are more likely to be rare, we next stratified variants based on MAF and applied a Madsen-Browning weighting scheme, which places more weight on rare variants and almost a zero weighting on more common variants. Restricting our analysis to 248 variants with MAF < 5% yielded an association between PCM1 and ASZ ($p = 0.033$; OR = 1.4; Table 1). Variants with MAF < 1% ($N = 202$) and variants with MAF < 0.1% ($N = 125$) were also associated significantly with ASZ ($p = 0.012$, OR = 1.4; and $p = 0.002$, OR = 1.6, respectively; Table 1). Upon examination of only non-synonymous variants with MAF < 0.1% ($N = 21$), we also observed a significant association between PCM1 and SSZ ($p = 0.020$, OR = 3.0; Table 2; Table 1).

Mindful of high false-positive potential of single-gene associations in complex traits, we obtained a replication sample of independent Danish ASZ (477 cases and 479 controls),

ascertained under the same phenotypic criteria as the discovery cohort and we sequenced PCM1. Similar to our discovery cohort, we did not observe an overall association between coding and noncoding PCM1 variants and ASZ ($p = 0.115$; Table 1). Given the significant genetic signal in non-synonymous variants with MAF < 0.1% from our discovery cohort, we focused on this subset of variants. Using SKAT-O, we found a similar enrichment of rare variants in replication cases ($N = 17$; $p = 0.006$, OR = 4.2; Table 3; Table 1). A meta-analysis of both the cohorts supported our observations further: we saw a significant association of ASZ ($N = 38$; $p = 0.001$, OR = 3.6; Table 4), suggesting that rare coding variation in PCM1 is associated with an ASZ subset.

Although the statistical results were encouraging, most of the association was driven by ultra-rare variants (Fig. 5a). We lacked statistical power to detect a significant association at the individual allele level. To address this limitation, we deployed a cilia in vivo complementation assay in zebrafish embryos[39,40]. We identified the sole zebrafish ortholog of PCM1 in the zebrafish genome, against which we designed two morpholinos (MOs): a translation-blocking (tbMO) and a splice-blocking (sbMO) morpholino (Supplementary Fig. 11a–c). First, we tested whether our zebrafish morphants could reproduce key brain anatomical pathologies in SZ. We focused on ventricle size, since this pathology is not only a consistent feature of the mouse model, but it is also one of the phenotypes observed reproducibly in multiple neuroanatomical imaging studies of some individuals with SZ[18,41]. Suppression of pcm1 induced an enlargement of ventricles in 3-day post-fertilization (dpf) embryos that could be grouped into three objective classes corresponding to severity ($p < 0.001$ for either MO against control, duplicated and scored blindly; Fig. 5b, c). Specificity of the observed phenotypes was corroborated by two independent lines of evidence. First, we found that ventricular volumetric defects could be rescued by the co-injection of the MO with human PCM1 mRNA ($p < 0.001$). These observations suggest that our findings are specific to the loss of pcm1 and that they are not due to toxicity from the MO.

We next used in vivo complementation of ventricular size with human mRNA to test systematically all 38 ultra-rare variants found in ASZ cases or controls from the discovery and replication cohorts. Assignment of an allele effect should enable us to assess the overall locus burden based on the aggregate number (and frequency) of pathogenic alleles in cases compared to controls. Injection cocktails consisting of the tbMO or sbMO and each of the PCM1 variants were co-injected into zebrafish embryos at the 1–4 cell stage and compared to the MO alone or MO plus WT human PCM1 mRNA (i.e., MO alone, MO + WT mRNA, MO + variant 1 mRNA, etc.; Supplementary Table 1). We found that we were unable to rescue the ventricular phenotype in 32% of the variants (12/38) that were present exclusively in cases, indicating that they were likely loss of function (LOF) alleles (Fig. 5c, d). Another 26% of variants (10/38) found in cases scored better than the MO alone, but were significantly worse than the WT rescue, suggesting that these alleles were hypomorphic (Fig. 5c, d). The remaining 21% of variant cases (8/38) scored similar to WT rescue, and thus were labelled them benign (Fig. 5c, d). 100% of variants (8/8) found exclusively in our controls, as well as two common variants, were scored as benign (Fig. 5c, d; Supplementary Table 1).

As a confirmatory test, we repeated the injection protocol on all alleles from the discovery cohort using a separate assay for phenotypic defects that have been observed in other ciliary and basal body mutants and morphants[42]. Phenotyping at the 11-somite stage reproduced convergent extension (CE) defects consisting of a shortened body axis, widening of the notochord, and tail extension defects that could be grouped into two objective classes corresponding to severity[39,40]. Importantly,

**Table 1 Summary of sequencing statistics.**

| Subset | N Variants | Unweighted p value | Odds ratio | 95% Confidence interval |
|---|---|---|---|---|
| Discovery cohort | | | | |
| All *PCM1 variants* | 293 | 0.068 | 1.149 | 1.115–1.184 |
| MAF < 5% | 248 | 0.033 | 1.358 | 1.247–1.487 |
| MAF < 1% | 202 | 0.012 | 1.373 | 1.140–1.664 |
| MAF < 0.1% | 125 | 0.002 | 1.642 | 1.158–2.386 |
| All non-synonymous variants | 21 | 0.02 | 3.007 | 1.213–8.708 |
| Replication cohort | | | | |
| All *PCM1 variants* | 278 | 0.115 | 1.048 | 1.002–1.097 |
| All non-synonymous variants | 17 | 0.006 | 4.179 | 1.213–8.708 |

**Table 2 Cohort 1: Discovery cohort, Gene-based association of PCM1 and ASZ using ultra-rare functional variants.**

| Subset | N variants (Cases) | N variants (Controls) | Unweighted p value | Odds ratio | 95% Confidence interval |
|---|---|---|---|---|---|
| All synonymous variants[a] | 16 | 5 | 0.200 | 3.007 | 1.213–8.708 |
| Benign | 5 | 5 | 0.876 | 1.000 | 0.296–3.377 |
| Hypomorph | 5 | 0 | 0.030 | 10.987 | 1.249–999 |
| Loss of function | 6 | 0 | 0.015 | 13.084 | 1.539–999 |

[a]Ultra-rare MAF< 0.1% (sequence kernel association test, SKAT-O).

**Table 3 Cohort 2: Replication cohort, Gene-based association of PCM1 and ASZ using ultra-rare functional variants.**

| Subset | N variants (Cases) | N variants (Controls) | Unweighted p value | Odds ratio | 95% Confidence interval |
|---|---|---|---|---|---|
| All synonymous variants[a] | 14 | 3 | 0.006 | 4.179 | 1.441–16.048 |
| Benign | 3 | 3 | 0.607 | 1.000 | 0.213–4.742 |
| Hypomorph | 5 | 0 | 0.015 | 11.003 | 1.249–999 |
| Loss of function | 6 | 0 | 0.007 | 13.181 | 1.539–999 |

[a]Ultra-rare MAF < 0.1% (sequence kernel association test, SKAT-O).

**Table 4 Meta-analysis of both cohorts, Gene-based association of PCM1 and ASZ using ultra-rare functional variants.**

| Subset | N variants (Cases) | N variants (Controls) | Unweighted p value | Odds ratio | 95% Confidence interval |
|---|---|---|---|---|---|
| All synonymous variants[a] | 30 | 8 | 0.001 | 3.598 | 1.76–8.22 |
| Benign | 8 | 8 | 0.666 | 1.000 | 0.38–2.642 |
| Hypomorph | 10 | 0 | 0.003 | 20.912 | 2.713–999 |
| Loss of function | 12 | 0 | 0.001 | 25.192 | 3.314–999 |

[a]Ultra-rare MAF< 0.1% (sequence kernel association test, SKAT-O).

20/21 alleles tested by both assays had the same qualitative score (benign or pathogenic), with one allele scoring as a hypomorph in the ventricle size assay but null in the CE assay (A1555G) (Supplementary Fig. 11d, e).

Informed by this functional annotation, we re-analyzed the human genetic data from the Danish SSZ cases and controls while considering only benign ultra-rare variants, hypomorphic ultra-rare variants, or LOF ultra-rare variants. While the benign variants were not significant ($p = 0.61$), the hypomorphic and LOF variants were associated significantly with ASZ ($p = 0.003$, OR = 20.9 and $p = 0.001$, OR = 25.2, respectively; Fig. 5d). A simpler binning of alleles into pathogenic relative to benign categories was even more significant ($p = 4.5 \times 10^{-5}$, OR = 44.6), aided by the increased overall number of alleles. Taken together, the distribution and incidence of non-synonymous variants in both of our cohorts and the functional modeling indicate that an

excess of pathogenic *PCM1* variants are present in cases with ASZ.

## Discussion
We have shown that loss of PCM1, a protein necessary for the organization of centriolar satellites[25] leads to structural ciliary defects in different regions of the postnatal brain, with concomitant behavioral defects that are not rescued with typical and atypical antipsychotic drugs. *PCM1* is expressed ubiquitously[43] (Supplementary Fig. 5d, e) and, given the critical role of this molecule, we expected widespread syndromic features. We speculate that the genetic background of these mice may exert a protective effect, especially since the importance of genetic background is a well-known factor in determining phenotypic expressivity in ciliopathies[44]. It will be important to analyze

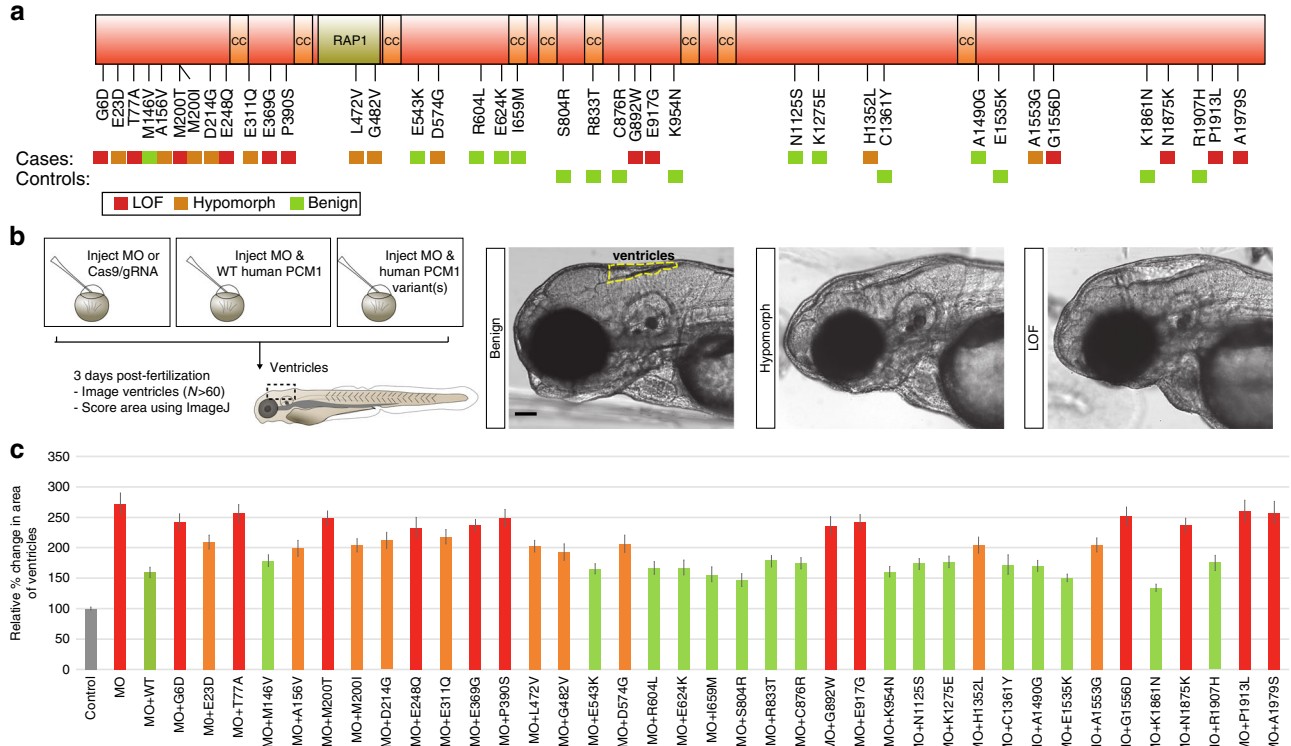

**Fig. 5 Enrichment of rare, pathogenic *PCM1* mutations in ASZ and functional modeling. a** Schematic depicting PCM1 protein and the location of protein interaction domains, and ultra-rare non-synonymous changes identified in both severe SZ cases and controls from the discovery and replication cohort. Colored boxes indicate loss of function (LOF), hypomorphic, or benign results from functional modeling in the zebrafish. **b** Schematic showing the functional modeling procedure in zebrafish embryos. 3 dpf, embryos imaged and ventricles quantified using ImageJ. Representative images for benign, hypomorphic, and LOF alleles are presented. Scale 100 μm (**c**) Normalized values for the average ventricular area of each variant discovered from severe SZ cases and controls are plotted relative to MO and MO + WT conditions. N fish/variant shown in Supplementary Table 1 with *p* values. Data as mean ± SD. LOF loss of function, CC coiled-coils domain, RAP1 rhoptry-associated protein 1 sequence, CI confidence interval, MO morpholino, WT wild-type. Detailed statistics in Table 1. Source data are provided as a Source Data File. Illustrations are original.

$Pcm1^{-/-}$ mice bred into different genetic backgrounds, as well as with mice in which null alleles for other ciliary and centriolar loci are introduced concurrently.

Our findings also reveal a potentially attractive model for exploring the pathophysiology of SZ, a neurodevelopmental disorder whose genetic susceptibility can be traced in some individuals to neuroanatomical defects[45]. For *PCM1*, we favor a model in which a gradual reduction in ciliary length reaches a threshold, at which time behavioral defects emerge. We postulate that a variety of other genetic and stochastic events, as well as environmental stressors, may alter the rate of ciliary degeneration, as well as spatial distribution of such defects in the CNS, and, thereby, alter the timing of illness onset.

Loss of *Pcm1* is associated with defects in responding fully to antipsychotic treatment. The observed reduction of D2R in striatal membranes is unlikely to explain fully the reduced responsiveness to the antipsychotic drugs tested. While the D2R is a major target for antipsychotic drugs[45], the pharmacologies of antipsychotics are complex. Our expression studies suggest that *Pcm1* mice may exhibit also other receptor-mediated signaling defects that merit further investigation. As such, the D2R phenotype and the ciliary length defects may represent two different endpoints of defects in microtubule organization affected by the loss of *Pcm1*. This possibility is consistent with the known microtubule-organizing role of PCM1 and other centriolar proteins[46].

We cannot exclude the possibility that the behavioral phenotype in the *Pcm1* mutant mice presents secondarily to the adult-onset lateral ventricle enlargement. For instance, it is known that

the cilia direct cerebrospinal fluid flow to ensure proper ventricle morphology[47]. In addition, the cilia are linked to congenital hydrocephalus[48]. Indeed, defective cilia during development frequently lead to severe hydrocephalus and multi-organ dysfunction incompatible with life[49–53]. As severe congenital hydrocephalus is an extremely dangerous condition, affected individuals may not be healthy enough to undergo proper psychiatric evaluations; however, apparent adult hydrocephalus has been reported to be comorbid with psychosis and paranoid delusions[54–56]. Others have proposed a CSF-forward model of SZ, but causality has not been tested explicitly[57]. *PCM1* LOF may represent such a scenario relevant to that model of cognitive decline, as the phenotypes present later in life. In future experiments, we plan to conditionally delete core cilia genes in postnatal brain cell types to formally address causality between the cilia integrity, lateral ventricle size, and cognitive decline.

Previous studies have shown an association of *PCM1* with SZ in some, but not all cohorts[8,12,13,58]. Our findings indicate that *PCM1* variants may contribute to a risk for SZ in specific subsets of individuals. For example, one study has suggested an association with orbitofrontal cortex gray matter defects[8]. Motivated by our previous findings showing a nonsense mutation in *PCM1* in human cases treated with clozapine, we used a SZ cohort at an extreme end of the phenotype (minimal responsiveness to three or more antipsychotics and committed to a psychiatric ward). Taken together with the anatomical and behavioral pathology in the $Pcm1^{-/-}$ mouse, our results indicate that *PCM1* may have a strong genetic effect at extremes of SZ. In this regard, both the functionally-informed sequencing of larger cross-sectional and

severely affected SZ cohorts will be required to delineate the precise contribution of this molecule to SZ. The complex and genetics underlying SZ, combined with the fact that affected individuals do not all respond to medication emphasizes the likelihood that the clinical presentation of SZ results from disparate underlying cell-biological mechanisms. As we continue to understand more clearly the basis for illnesses, such as SZ, we move closer to a future where genetic diagnoses can advance our efforts to provide patient-specific treatments.

## Methods

**Generation of knockout mice.** Gene-trap embryonic cells (parental cell line, E14Tg2a.4: AZ0079) were purchased from the International Gene Trap Consortium and implanted into C57BL/6J pseudo-pregnant dams. Mice were born and maintained at the Johns Hopkins University before re-derivation at Duke University on a C57BL/6J background (at least three generations backcross). Genotyping was performed using primers that span the insertion (PCM1_F: CTGCACT TGAGTGTTCAGTAC; PCM1_R: GCTCAACATTGGCAGATCTTT) and an insert-specific pair (Bgeo_F: GCGAATACCTGTTCCGTCATA; Bgeo_R: GAGC GTCACACTGAGGTTTTC). The mm10:insert junction was mapped by first amplifying it using PCM1_F and Bgeo_R, and then using the PCM1_F primer for Sanger sequencing. All procedures were approved by the Duke University Institutional Animal Care and Use Committee and were conducted in accordance with NIH guidelines for the care and use of laboratory animals. Mouse N numbers were chosen based on past experience with the assays. Mice were randomized for experimentation, and only included into experimental groups after genotyping. Both male and female mice were used.

**Preparation of mice and magnetic resonance imaging.** Fixation of tissues in P90 $Pcm1^{-/-}$ mice and littermates was performed by transcardial perfusion with 0.9% saline to flush blood from the brain followed by 10% Neutral Buffered Formalin (NBF) containing 10% (50 mM) Gadoteridol (Bracco Diagnostics, Monroe Township, NJ) to reduce the spin lattice relaxation time (T1) of the tissue, and to enhance the magnetic resonance signal. Mouse heads were subsequently stored in 10% NBF for 24 h before transfer to a 0.1 M phosphate buffered saline solution containing 0.5% (2.5 mM) Gadoteridol at 4 °C for 5–7 days to rehydrate the tissue. Extraneous tissue around the cranium was removed prior to imaging, and specimens were placed in MRI-compatible tubes, immersed in liquid fluorocarbon (Fomblin; Solvay Solexis; Fisher Scientific, Hampton, NH) for susceptibility matching and to prevent specimen dehydration. As previously described, specimens were scanned within the cranium to avoid tissue damage and possible distortions, and they remained in the same position in the magnet during the image acquisition protocol[58]. Imaging was performed using a 7 Tesla small animal MRI system (Magnex Scientific, Yarnton, Oxford, UK) equipped with 650 mT m$^{-1}$ Resonance Research gradient coils (Resonance Research, Inc., Billerica, MA), and controlled with an Agilent Direct Drive console (Agilent Technologies, Santa Clara, CA) running VNMRJ 4.0 (Agilent Technologies). Radiofrequency (RF) transmission and reception were achieved with a 13 mm diameter solenoid coil made in-house. Brain specimens were imaged with a 3-D multi-gradient echo sequence with repetition time 50 ms, echo times 4.24 ms, echo spacing 6.28 ms, using 4 echoes, with a flip angle a = 60°, and bandwidth 62.5 kHz, sampling with a matrix of 480 × 240 × 200, over a field of view 24 × 12 × 10 mm, yielding an isotropic resolution of 50 μm. Images were analyzed using ITK-SNAP[59] and ImageJ[60]. No inconsistent variation was detected between the groups.

**Immunohistochemistry, histology, and imaging.** Upon perfusion, mouse heads were either placed in 30% sucrose overnight before cryosectioning or were embedded in 4% low melting-point agarose and sectioned at 100 μm using a vibratome. Alternatively, we prepared formalin-fixed, paraffin-embedded 7 μ sections for histology. Brain sections were stained with the following antibodies from Santa Cruz Biotechnology (AC3, sc-588 1:500; MCHR1, sc-5534 1:100; SSTR3, sc-11617 1:500), Abcam (Arl13b, ab83879 1:500), Sigma-Aldrich (PCM1, HPA023370 1:200), and Millipore. Alexa-fluor secondary antibodies [anti-rabbit, A-11008; anti-goat, A-11055; 546(anti-rabbit A-11035; anti-goat A-11056)] were purchased from Life Technologies. Images were collected on a Zeiss 780 confocal microscope and processed using Imaris software (www.bitplane.com), Adobe Photoshop, Illustrator, and ImageJ. Histology was performed in the Northwestern University Mouse Histology and Phenotyping Laboratory using standard methods for Cresyl violet or hematoxylin and eosin staining. Neuronal density was determined after Cresyl violet staining by counting Nissl substance-positive nuclei per area in section-matched tissue. $Pcm1$ RNA in situ and expression data from the Allen Brain Atlas (https://mouse.brain-map.org/gene/show/18302) were used. No inconsistent variation was detected between the groups.

**Behavioral tests.** Adult male and female littermate mice were housed under controlled temperature and humidity conditions on a 14:10 h light:dark cycle with light onset at 0800 h. Food and water were provided *ad libitum*. Except for

anhedonia which was conducted during the dark cycle, all behavioral tests were conducted between the hours of 1000 h and 1600 h. Mice were tested in the following procedures as described below: open field, novel object recognition memory, Morris water maze, sucrose preference, PPI, and catalepsy. Animals were evaluated in up to three tests, with a minimum of 10–14 days imposed between tests. Mice that received PCP in the open field were a separate cohort of animals. Animals that were examined in PPI and received either haloperidol or clozapine treatments were examined only once, or if used twice, they had received vehicle 2 weeks prior to drug treatment in PPI. In no case was an animal examined in PPI with both antipsychotics.

**Drugs used.** Phencyclidine (PCP), haloperidol, and clozapine were purchased from Sigma-Aldrich (St. Louis, MO). PCP was dissolved in sterile water (Butler-Schein Animal Health, Dublin, OH); haloperidol (Bio-Techne, Minneapolis, MN) was dissolved in sterile-filtered DMSO (Sigma-Aldrich, St. Louis, MO) and diluted in sterile water (Hospira Inc., Lake Forest, IL). For haloperidol, the final concentration of DMSO did not exceed 0.05% of the injected volume. Clozapine (Bio-Techne) was dissolved in 0.2% glacial acetic acid with sterile water. The glacial acetic acid did not exceed 0.01% the final injected volume.

**Open field.** Spontaneous and phencyclidine (PCP) stimulated locomotor activities were evaluated in a 21 × 21 × 30 cm Plexiglas open field (Omnitech Inc., Columbus, OH) at 340 lux. Spontaneous activity was monitored over 60 min. In the PCP study, mice were habituated to the open field for 60 min, injected with the vehicle or 4 or 6 mg kg$^{-1}$ PCP, and returned immediately to the open field for an additional hour. Activity was monitored by infrared diodes interfaced to a computer running Fusion software (Omnitech). Locomotion was scored as distance (cm) traveled in 5-min segments or as cumulative distance traveled in the first hr after injection.

**Prepulse inhibition.** For PPI, mice were administered the vehicle, haloperidol, or clozapine. Twenty-five min after injection, mice were placed into the PPI apparatus (SR-LAB chambers; San Diego Instruments, San Diego, CA) and acclimated for 10 min before the start of testing. Testing was composed of three types of trials: pulse-only, prepulse + pulse, and null trials. The pulse-only trials consisted of a single 40 ms burst of 120 dB white-noise startle stimulus. Prepulse + pulse trials were composed of a 20 ms burst of white noise which was 4, 8, or 12 dB above the white-noise (62 dB) background followed 100 ms later by the 120 dB startle stimulus. Null trials consisted of the white-noise background alone. Each PPI test was comprised of 42 trials, which began with 6 startle-only trials, and completed with 6 startle only trials. The remaining 30 trials were delivered between these two sets of startle-only trials and were randomized equally between 6 startle-only, 6 null, and 6 of each type of prepulse + 4, 8 or 12 dB prepulse + pulse trials, and finishing with 5 startle-only trials. PPI responses were calculated as % PPI = [1–(pre-pulse trials per startle-only trials)]*100. The pulse only and null trials were expressed as arbitrary units (AU).

**Novel object recognition memory testing.** Mice were acclimated to the test chamber (40 × 22 × 18 cm) for 2 days before the start of testing. On test day 1, mice were exposed to a pair of identical objects (~3.5 cm square) for 5 min. Thirty min following training, one of the identical objects was replaced with the first novel object for a short-term memory (STM) assessment. Twenty-four hour after training, a new novel object replaced one of the training objects for the assessment of long-term memory (LTM). All animals in each test were tracked for activity and object interaction using Ethovision 11 software (Noldus Information Technologies, Leesburg, VA). From the 3-point tracking profiles (nose-body center-trail) the duration and frequency of the nose contacts within 2 cm of each object were scored. The preference score for the novel object was calculated as: [(time spent with the novel object - time spent with the familiar object)/total time spent with both objects].

**Morris water maze (MWM).** Mice were handled and acclimated to standing in water for 1 week before testing in a room that was different from that used for the novel object recognition memory test. Mice were moved to the water maze room 18 h before the start of testing. The water maze (120 cm diameter) was a stainless steel pool divided into four quadrants with a platform hidden 1 cm below the surface of opaque water in the northeast (NE) quadrant. Animals were trained daily to swim to the hidden platform over 6 consecutive days. Each training day was comprised of four trials given in pairs, with each pair of trials separated by 60 min. An acquisition trial ended when the mouse found the hidden platform or it was terminated after 1 min of searching without success. For probe tests the hidden platform was removed from the pool and it was conducted 1 h following completion of the final test trial on days 2, 4, and 6. Each probe test was 1 min in duration. Performance of the mice on all trials was assessed by swim distance (cm) to the hidden platform from tracking profiles created by Ethovision 11 (Noldus Information Technologies, Leesburg, VA). For probe tests the total swim distance (cm) by each animal in each quadrant of the maze was measured.

**Anhedonia**. Mice were housed in separate home cages and were presented with two separate bottles of water every 24 h for 3 consecutive days to record baseline consumption (W-W comparison). Following training, one of the water bottles was replaced with one containing a sucrose (0.5, 1, or 2%) or quinine (15 or 30 mM) solution. Each concentration of sucrose or quinine was presented for 3 consecutive days with the positions of the water and sucrose or quinine bottles switched daily. Several days with a single water bottle were interposed between sucrose or quinine concentration tests. For each test, sucrose or quinine, the lowest concentration was presented first and progressed to the highest concentration. Quinine testing was conducted after sucrose testing was completed. Preference scores were calculated as: [(volume of sucrose or quinine consumed – volume of water consumed)/total volume of liquid consumed]. For each concentration of sucrose or quinine, the average preference score over the 3 test days was calculated for each animal, including the average total volume of fluid consumed.

**Catalepsy**. This response was assessed with the horizontal bar test. A baseline (time 0) response was obtained for each animal. Subsequently, mice were injected with different doses of haloperidol and re-tested for catalepsy 1 h after injection. In each case, testing started when the forepaws of the animal were placed on the bar and the latency to remove the paws was scored. The data are presented as time spent in catalepsy with a 60 s cut-off.

**Olfaction**. This was assayed with the hidden cereal test. After a 2-day familiarization phase with cereal freely available, mice were fasted overnight. Subsequently, they were placed in a new cage with clean bedding and a cereal morsel buried 1 cm below the surface. They were timed until they begin to consume it.

**Behavioral statistics**. All behavioral data are presented as means ± standard errors of the mean (SEMs) and were analyzed with SPSS 24 (IBM SPSS Statistics, Chicago, IL). Independent measures *T*-tests were used to evaluate single behaviors comparing the two genotypes, ANOVA was used analyze single responses with multiple independent variables (genotype, treatment). Repeated measures ANOVA (RMANOVA) were used to assess responses measured over time or in tests with several functional dependent variables obtained simultaneously. In all cases, Bonferroni corrected pair-wise comparisons were used for *post-hoc* analyses and $p < 0.05$ was considered statistically significant.

**RNA sequencing and qRT-PCR**. For RNAseq, tissues from various brain regions of $Pcm1^{-/-}$ and littermate WT mice were dissected and snap-frozen on the same day. Using the RNeasy mini kit (Qiagen, Hilden, Germany), RNA was isolated and both quantity and quality were determined using the Agilent RNA ScreenTape assay and TapeStation. RNA sequencing libraries were prepared using KAPA stranded mRNAseq kits and sequenced on an Illumina HiSeq platform. RNAseq data were processed using the TrimGalore toolkit (www.bioinformatics.babraham.ac.uk/projects/trim_galore), which employs Cutadapt to trim low quality bases and Illumina sequencing adapters from the 3′end of the reads. Only reads that were 20 nt or longer were subjected to further analysis. Reads were mapped to the GrCm38r73 version of the mouse genome and transcriptome using the STAR RNAseq alignment tool[61]. Reads were kept for subsequent analysis if they mapped to a single genomic location. Gene counts were compiled using the HTSeq tool[62]. Only genes that had at least 10 reads in any given library were used in subsequent analysis. Normalization and differential expression were performed using the EdgeR Bioconductor package[63] with the R statistical programming environment. Gene Set Enrichment Analysis[64] was used to identify significant pathways and gene ontologies that were being differentially regulated between conditions. For qRT-PCR, RNA preparations were made from sectioned paraffin-embedded tissue using the RecoverAll Total Nucleic Acid isolation kit for FFPE (ThermoFisher AM1975) and used with the following $Pcm1$ primers: F: AAATCCTTGCCAGAGATCCTCA R: CAGCTGCTCCTGCTGTTGTA, and $Gapdh$ primers F: AGGTCGGTGTGAACGGATTTG, R: TGTAGACCATGTAGTTGAGGTCA. Amplification was measured using the Power SybrGreen PCR Master Mix (Applied Biosystems Cat#4367659). Results are normalized to $Gapdh$.

**Immunoprecipitation**. HT22 hippocampal cells and mouse tissues were homogenized in lysis buffer containing Complete Protease Inhibitors (Roche). Antibodies against FLAG (Sigma, F3165 1:5000), HA (Sigma, H6908 1:1000), D1R (Santa Cruz, sc-1434 1:1000), D2R (Millipore, RAB5084P 1:1000), and PCM1 (Santa Cruz Biotechnology, sc-67204 1:500) were used in the immunoprecipitation and/or immunoblotting assays. HRP-conjugated secondary antibodies [GE Healthcare (cytiva), anti-Goat (NA31IV), anti-Rabbit (NA934), anti-Mouse (NA931)] and Super-Signal West Pico Chemiluminescent Substrate (Thermo Scientific) were used to develop the antibody signal. Image acquisition was performed using a ChemiDoc XRS + imaging system (Biorad) and images were assembled using Adobe Photoshop and Illustrator.

**D2R saturation binding assay on striatal membranes**. Striatal brain tissue from $Pcm1^{-/-}$ or WT littermate mice was dissected and placed on dry-ice before homogenization on the same day using a Teflon-glass homogenizer in 3 ml of lysis

buffer containing 50 mM Tris–HCl (pH 7.4), 120 mM NaCl, 1 mM EDTA, and a cocktail of protease inhibitors at 1:1000 dilution. The homogenate was centrifuged at 1000 rpm for 10 min at 4 °C to remove tissue debris and nuclei. The resulting supernatant was centrifuged twice at 40,000 x g for 20 min at 4 °C and the final pellet was suspended in assay buffer containing 50 mM Tris–HCl (pH 7.4), 120 mM NaCl, 5 mM KCl, 2 mM CaCl₂, and 1 mM MgCl₂. [³H]-Raclopride (Perkin-Elmer), a selective D2R antagonist was used for D2R saturation experiments. Membranes (50 μl; ~1 μg μl⁻¹ final protein concentration) were incubated with 6 increasing doses of [³H]-raclopride (range: 1.6–52 nM at 50 μl). The reactions were conducted at room temperature for 1 h in a total volume of 100 μl of assay buffer. Nonspecific binding was measured using nonradioactive 100 μM raclopride in parallel assay tubes and subtracted from total binding to obtain specific [³H]-raclopride binding. The incubations were terminated by rapid filtration over Brandel GF/C glass fiber filters washed with ice-cold buffer containing 50 mM Tris–HCl. The filters were pre-soaked in 0.1% polyethyleneimine (Sigma, P3143) at 4 °C for at least 1 h before being incubated overnight in 4 ml high-flash point scintillation cocktail (Lefko-Fluor). Radioactivity content was counted using a liquid scintillation counter. The experiment was repeated three times with three animals/genotype; the distribution of outcomes (Bmax and Kd) were assessed and outliers (>3 standard deviations from the mean) were removed. ANOVA with post-hoc Bonferroni correction was performed to analyze differences in means between WT ($N = 8$) and $Pcm1^{-/-}$ ($N = 9$) groups, controlling for experimental batch using the removeBatchEffect function within the limma library in R. No inconsistent variation was detected between the groups.

**Human patient recruitment**. The Danish cohort was recruited by the Danish Psychiatric Biobank from the psychiatric wards in Copenhagen. All individuals had been diagnosed with SZ according to ICD-10 (F20) at repeated hospital admissions. An experienced research and consultant psychiatrist verified the reliability of the clinical diagnoses using OPCRIT semi- structured interviews[37]. Approximately 85% of the individuals were ethnic Danish (individuals and both parents were born in Denmark), while ~15% of individuals had one parent who was Caucasian and born outside Denmark in another Northern European country -- primarily in Sweden or Norway, secondarily in Germany, the Netherlands, England or France. The healthy control subjects were selected anonymously among 72,000 unpaid voluntary blood-donors from the Blood Donor Corps in the Copenhagen area. The Donor corps included ~5% of the adult population aged 18–65 years. Apparent behavioral abnormality was an exclusion criterion and all individuals stated that they felt completely healthy with a possibility to discuss any health-related issues with a physician. A control subject of equivalent ethnic background was matched to each individual on gender, and year and month of birth. Individuals were recruited according to a protocol approved by the Danish Scientific-Ethical Committee (No 01-024/01) and the Data Protection Agency (No 2001-54-0798).

**Sequencing and statistical analyses**. Sequencing was performed on the discovery cohort using an Illumina HiSeq sequencing platform (Illumina) and bclToFastq 1.8.3 (Illumina) was used to generate FASTQ files from bcl (base-calling) files. Sequencing on the replication cohort was performed on an Ion Torrent. Sequences were aligned to the hg19 reference genome using the Burrows-Wheeler Aligner (BWA[65]). The BamUtil package was employed to assess the mapping quality and duplication rates of aligned reads, and Picard and SAMtools were used to remove unmapped read and PCR duplicates. Ten samples were experimental outliers (mapping rate < 95%) and were removed. The GATK[66] multi-sample Haplotype-Caller was used to make genotype calls[67] and variant annotation was performed with ANNOVAR[68]. VCF files were converted to PLINK[69] using PLINK/SEQ and all subsequent sample and variant assessments, including MAF calculations, were performed in PLINK. Identity-by-state (IBS) estimates were generated and eight individuals were identified as either duplicates ($N = 5$) or first-degree relatives ($N = 3$) and were excluded. To determine if population substructure existed among the study samples, we performed PCA using linkage disequilibrium (LD)-pruned common variants (MAF > 5%) with the smartpca program in EIGENSOFT[70]. Genome-wide data from the International HapMap project for the following populations were analyzed with the study samples: CEPH (Utah residents with ancestry from northern and western Europe; CEU); Yoruba in Ibadan, Nigeria (YRI); Japanese in Tokyo, Japan (JPT); and Han Chinese in Beijing, China (CHB). Because the only available genotype data for all study samples were from this targeted sequencing panel, there were only 5,754 variants in genes other than $PCM1$ which overlapped the study samples and the HapMap samples. Principal components were plotted using R (www.r-project.org) and inspected for outliers (Supplementary Fig. 10). Nine study samples (1 control and 8 cases) were outliers and were removed. We employed the optimal gene-based sequence kernel association test (SKAT-O) to test for association between $PCM1$ and SSZ. SKAT-O includes both burden tests and the original SKAT test to maximize power, as either may be more powerful depending upon the nature of each specific association. SAS v9.4 was used to generate FDR $q$-values, as well as odds ratios and 95% confidence intervals using penalized maximum likelihood for low cell counts (Firth's method). Human developmental $PCM1$ expression data were retrieved from the Allen Brain Atlas of the human developing brain[27].

**Zebrafish injection and analyses**. All zebrafish experiments were approved by the Duke University Institutional Care and Use Committee (Protocol A154-18-06). Using reciprocal BLAST, we identified the annotated zebrafish ortholog for *pcm1* with high amino acid similarity between human and zebrafish (64% similarity and 51% identity). A splice-blocking (SB: 5′-CATCTTAAACATCGCTCGTACCAGT-3′) and translation-blocking (TB; 5′-AGTGCCACCCGTTGCCATGATGAAC-3′) morpholinos (MOs) against exon 6 and the initiation exon of the zebrafish *pcm1* gene, respectively, were designed by Gene Tools. One to four cell stage Casper, Ek, Ek/AB embryos were injected with a 1 nl cocktail at a concentration of 12 ng μl$^{-1}$ and embryos were incubated at 28 °C for 72 hpf for ventricular area assessment and at 23 °C to a 11-somite stage (±one somite) for CE evaluation before live scoring. Validation of MO activity was determined using RT-PCR (Supplementary Fig. 11a, b) with the following primers: dr_β_actin_F: GAGAAGATCTGGCATCACACC; dr_β_actin_R: AGCTTCTCCTTGATGTCACG; dr_pcm1_Ex4F: AGAGGTTGG-GATTGGAGGAT; dr_pcm1_Ex7R: AAGTAATGCTCCTGCCAGACA. CRISPR results were generated from three CRISPR guide RNA oligonucleotides that were synthesized, annealed and ligated into pCS2. The CRISPR oligonucleotide sequences are the following, Ex2.1: CCTGCTCCAGGGGTAGACTC, Ex2.2: TAGGATTTCTCCAGAGTCTACCCC, and Ex3: TGTTCCTTGCCCCCAGGTG. Sufficient animal numbers were used according to our previous experience (>30 per allele). Zebrafish were randomized with controls. Quantification was performed blinded to genotype. No inconsistent variation was detected between the groups.

## Data availability

The RNAseq datasets are available at GEO, accession GSE145915. In situ data from the Allen Brain Atlas Allen Brain Atlas [https://mouse.brain-map.org/gene/show/18302]. The publicly-available hg19 human reference genome, as well as the GrCm38r73 mouse genome were used for mapping sequencing data. The human sequencing data is available upon reasonable request and an application for use from The Danish Schizophrenia Registry [https://www.danishhealthdata.com/find-health-data/Den-Nationale-Skizofrenidatabase], and upon approval by the Danish Data Protection Agency. Source data are provided as a Source Data file and all other materials are available upon reasonable request. Source data are provided with this paper.

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

## Acknowledgements

We thank Dr. Edwin Oh for initiating the project and contributing to several experiments, E. Golden, K. Childs, and K. Bendt for assistance with mouse husbandry, J. Crowley and R. Nonneman for assistance with haloperidol dosing, and T. Rhodes and C. Means for conducting behavioral experiments and pharmacology in the Mouse Behavioral and Neuroendocrine Analysis Core Facility at Duke University. Histology services were provided by the Northwestern University Mouse Histology and Phenotyping Laboratory, supported by NCI P30-CA060553 awarded to the Robert H. Lurie Comprehensive Cancer Center. Some of the equipment and software used in the behavioral studies were purchased with a grant from the North Carolina Biotechnology Center. Magnetic resonance imaging was performed at the Duke Center for In Vivo Microscopy and the NIH/NIBIB National Biomedical Technology Resource Center (P41 EB015897). This work was supported by NIH grants: 5T32DK108738-04 (T.O.M.), U54 HG002373 (R.A.G.), Silvo O. Conte Center grant MH084018 (N.K. and A.S.). N.K. holds the Valerie and George D. Kennedy Professorship in Human Genetics. Investigators were blinded to genotype.

## Author contributions

Conceptualization: N.K. Methodology: T.O.M., M.K., R.M.R., R.G., A E.A.-K., T.W., A.S., and W.C.W. Investigation: T.O.M., M.E.G., M.K., R.M.R., S.M., Y.B., S.C.B, K.L.S., J.S., T.F.H, D.M.M., L.B., P.S., and A.E.A.-K. Resources: R.G., A.S., W.C.W., T.W., and N.K. Writing – Original Draft: T.O.M. and N.K.. Writing – Review and editing: T.O.M., A.S., W.C.W., and N.K. .

## Competing interests

N.K. holds significant stock in Rescindo Therapeutics. The remaining co-authors declare no competing interests.
