## [Peer Review File · Nature Communications]

Reviewers' comments:

Reviewer #1 (expertise: neurodevelopment, cilia) (Remarks to the Author):

In this work, Edwin Oh and colleagues study the mutant phenotype of Pcm1 gene trap mutants and show convincing anomalies of cilia and some brain anomalies. They then try and link this to human schizophrenia-associated variants and to validate some variants in zebrafish using morpholinos. Although there is a lot of work involved in this work, it is hard to avoid the impression that authors have been carried somewhat astray by their desire to link to human pathology. Although the aim is laudable, this approach leads them to cut corners when analyzing the mouse phenotype, and to focus on fancy technology rather than on strong neuroanatomical data that are quite lacking.

A good example is the neuroanatomy shown in Fig.1. Authors focus on MRI data and show some preliminary histological images only in supplement. Obviously, the other way, although arguably less glamorous, would be much more useful. From Supplemental figure 1, it is quite evident that Pcm1 mutants have an abnormal hippocampus. Although section levels are not perfectly matched, the mutant DG is blunted, CA3 looks almost of normal size and CA1 is clearly diminutive. Those anomalies may be important and directly relevant to ciliary anomalies, given that cilia are known to regulate DG adult neurogenesis. That hippocampal phenotype should be investigated further using for example some Timm staining, some DiI tracing, crosses with Thy1 mice or other simple techniques. A similar comment can be made about cortex, where authors describe some markers and conclude that the cortex is unremarkable; same remark about striatal anomalies.

It is also very standard knowledge that the centrosome regulates neurogenesis. Authors do not consider this. They may have a good reason; for example, Pcm1 may not be expressed in ventricular zones. However, I did not find relevant information on Pcm1 expression in *euexpress*. Therefore, a good analysis of Pcm1 expression using ISH or IHC at E14, P0 and adult would be a valuable addition to the data.

This being said, that Pcm1 mutant has very interesting cilia anomalies and most probably neuroanatomical defects that authors failed to appreciate. This raises several interesting cell biological questions. Previous papers mentioning Pcm1 were mostly carried out *in vitro*. Therefore, a careful analysis of the mutant phenotype would lead to very significant contributions and progress in understanding the role of that protein in development and in adult.

The analysis of the mutant phenotype is so preliminary that the rest of the paper and the hard work in RNA seq, analysis of variants in human families and work in zebrafish, by not being really based on strong phenotypic traits to guide them, conveys the unavoidable impression that authors do not follow their data but tend to have some *a priori* in terms of human diseases.

Reviewer #2 (expertise: genetics/genomics brain development, mouse models) (Remarks to the Author):

In this study, Oh et al. investigate the *in vivo* functions of PCM1, a gene involved in cilia biogenesis. They discover that the loss of PCM1 leads to numerous behavioral phenotypes in adult mice that are relevant to symptoms of schizophrenia, such as hyperactivity, impaired prepulse inhibition, anhedonia, and impaired spatial learning and memory. Furthermore, they find that progressive ciliary defects develop postnatally in mice lacking PCM1, concurrent with the onset of several behavioral phenotypes. Transcriptomic profiling shows the dysregulation of GPCR and amine pathways, although the significance of these changes with respect to disease is unclear. Through the sequencing of Danish cohorts, the authors also link schizophrenia cases with ultra-rare variants of PCM1 and have tested their pathogenicity in an *in vivo* complementation assay in zebrafish. Overall, this is a potentially important study, revealing an interesting role for PCM1 in mouse neural development and an association between rare PCM1 variants and schizophrenia,

although the disease-causing mechanisms involving this gene remain largely unknown. The authors draw numerous conclusions based on loosely-connected evidence, such as alteration of D2R and catalepsy difference upon Haloperidol treatment, and multiple technical details are missing, making it difficult to evaluate the validity of the presented data.

Specific comments:

1. The authors state that "Pcm1^{-/-} mice were hyper-responsive to PCP." However, Figure 2C does not show that Pcm1^{-/-} mice demonstrate an over-exuberant response in comparison to wild type. To draw this conclusion, the authors will need to show comparisons of vehicle- and PCP-injected wild type and Pcm1^{-/-} mice at various doses. Furthermore, the authors did not discuss why they chose PCP as opposed to the better-tolerated (and more commonly used) ketamine, which provides a more specific NMDAR antagonism. Finally, would the hyperactivity phenotype be a significant confound to their PPI measurement?

2. The authors do not mention that PCP can also act as a 5-HT reuptake inhibitor, and that haloperidol and clozapine are also antagonists of certain subunits of the 5-HT receptor thereby potentially affecting the negative and cognitive symptoms associated with SZ by altering serotonin signaling. The link to amine pathway signaling in their RNAseq data further encourages the examination of this receptor group within Pcm1^{-/-} mice and their role in centriolar protein-associated SZ cases. It would be interesting to assess some of these phenotypes using a higher dose of haloperidol (>2.6mg/kg) to antagonize the 5-HT receptor to significant levels.

3. The authors state that "Subsequent to validation of the RNAseq-derived misregulation of D1R and D2R by qRT-PCR in Pcm1^{-/-} mice..." However, there is no reference to this data in the paper and it does not appear to be included. The authors also state in multiple places, including the abstract, about aberrant localization of D2Rs. Where is the data that show mis-localization of D2Rs?

4. Suppl. Fig. 1D is of poor quality. How can a conclusion be drawn from this figure? How do the authors know the staining is specific and meaningful?

5. Figure 1B presents a rather important finding, but additional measurements, such as brain size, brain weight, thickness of cortical layers, need to be carefully measured and presented.

6. Figure 3 also represents a key finding, but how are bulbous cilia measured and quantified? Representative images at P90-120 should be included as well. Similarly, how is the data for Suppl. Fig. 7 defined and quantified?

7. Progressive appearance of ciliary defects and progressive development of behavioral phenotypes are two key findings revealed by this study. But, the progressive nature of behavioral phenotypes is not well conducted. Other than PPI, locomotion, anhedonia, and learning and memory tests should be included for juvenile or prepubescent mice to corroborate the conclusion.

8. Is there any in vivo evidence supporting the association of D2R with PCM1?

9. How was catalepsy measured and quantified?

Reviewer #3 (expertise: psychiatry genetics) (Remarks to the Author):

The manuscript provides evidence for the involvement of genetic variation in the pericentriolar material 1 (PCM1) gene in susceptibility to schizophrenia. The gene has previously been reported to be associated with the disease but these findings did not show what would now be considered to be strong evidence for involvement in schizophrenia. There was also prior evidence that reduced expression of PCM1 led to impaired cortical migration.

The manuscript draws on evidence from sequence analysis of 880 severely affected schizophrenia subjects and 863 control subjects from Denmark. This sequencing identified a number of rare and "ultra-rare" variants. In aggregate these variants showed modest evidence for association with

schizophrenia. However, by modelling the ability of the mutations to recover a phenotype in zebra fish embryos the authors were able to discriminate between those genetic variants that were considered to be benign and those that were considered to be "pathogenic". Using this categorisation of the variants and selecting only the pathogenic variants the authors were then able to show evidence for association with schizophrenia risk ($p=4.5 \times 10^{-5}$, OR=44.6). The authors also present compelling and interesting data that a PCM1 mutant mouse shows brain structural changes similar to those reported in a PCM1 association study in humans. They also present detailed pharmacological, behavioural and gene expression data that suggest that ablation of PCM1 has profound impacts on processes thought to be linked to schizophrenia.

I believe that the manuscript provides important new evidence implicating the PCM1 gene in schizophrenia risk. The multiple lines of high quality data do not in isolation, present definitive evidence that the gene is involved in schizophrenia. However, the evidence from the manuscript combined with the prior findings provides good evidence that this gene may be involved in schizophrenia risk.

The authors should however provide more details on the human genetic variants that they have identified. They do not provide the genomic locations of the variants nor the nucleotide changes. It would also be useful to know whether these variants have been reported before in GnomAD (<http://gnomad.broadinstitute.org/>) or in the ExAC samples and in the non-psych ExAC samples (<http://exac.broadinstitute.org/>).

As stated above, the reported evidence for schizophrenia risk from combined analysis of the pathogenic variants is ($p=4.5 \times 10^{-5}$, OR=44.6). The authors should comment as to whether this finding exceeds a genome wide threshold of significance and/or indeed what the false positive rate would be in context of this form of analysis.

Response to Reviewer Comments

Reviewer #1

In this work, Oh and colleagues study the mutant phenotype of Pcm1 gene trap mutants and show convincing anomalies of cilia and some brain anomalies. They then try and link this to human schizophrenia-associated variants and to validate some variants in zebrafish using morpholinos. Although there is a lot of work involved in this work, it is hard to avoid the impression that authors have been carried somewhat astray by their desire to link to human pathology. Although the aim is laudable, this approach leads them to cut corners when analyzing the mouse phenotype, and to focus on fancy technology rather than on strong neuroanatomical data that are quite lacking.

We thank the reviewer for a thoughtful review and for the positive comments. We pride ourselves on using several lines of evidence and multiple organisms to test our hypotheses. We regret to have left an impression that the mouse data was insufficient, therefore, we have invested upwards of two years in generating substantial additional data.

A good example is the neuroanatomy shown in Fig.1. Authors focus on MRI data and show some preliminary histological images only in supplement. Obviously, the other way, although arguably less glamorous, would be much more useful. From Supplemental figure 1, it is quite evident that Pcm1 mutants have an abnormal hippocampus. Although section levels are not perfectly matched, the mutant DG is blunted, CA3 looks almost of normal size and CA1 is clearly diminutive. Those anomalies may be important and directly relevant to ciliary anomalies, given that cilia are known to regulate DG adult neurogenesis. That hippocampal phenotype should be investigated further using for example some Timm staining, some Dil tracing, crosses with Thy1 mice or other simple techniques. A similar comment can be made about cortex, where authors describe some markers and conclude that the cortex is unremarkable; same remark about striatal anomalies.

In the revised manuscript, we present extensive new analysis, leveraging our existing MRI data and new histology, which is now presented first. The new data can be found throughout Figure 1 and new Figure S2. We had similar thoughts on the hippocampi, based on their appearance in two dimensions, so we were surprised to find that the hippocampus sizes in fact were not different, and the cellular layer thickness had no appreciable differences. However, the morphology of the hippocampi in three-dimensional space was altered relative to WT, an observation that would be extraordinarily difficult to discern using histology alone. These data are now shown in Figure S2. Per the reviewer's suggestion, we did attempt TIMM staining, but were not able to obtain reliable results. We look forward to crossing in transgenic reporter mice in future neuro-connectivity-based projects.

It is also very standard knowledge that the centrosome regulates neurogenesis. Authors do not consider this. They may have a good reason; for example, Pcm1 may not be expressed in ventricular zones. However, I did not find relevant information on Pcm1 expression in eurespress. Therefore, a good analysis of Pcm1 expression using ISH or IHC at E14, P0 and adult would be a valuable addition to the data.

We thank the reviewer bringing up this prior data. Indeed, Pcm1 is expressed in the ventricular zone at E14¹ in mice. Therefore, this comment highlights an important feature of our mouse, in that they show no overt developmental phenotype. Deciphering how the centrosome can contribute to neurodegenerative phenotypes, as opposed to neurodevelopment, remains an active interest in our laboratory. We have also now included previously unreported adult mouse expression and RNA-*in situ* data for *Pcm1*, which can be found in Figure S5. We also noted no clear expression change in human *PCM1* throughout development, which we also plot in Figure S5.

1. **Figures 4e and f.** Ge, X., Frank, C. L., Calderon de Anda, F. & Tsai, L. H. Hook3 interacts with PCM1 to regulate pericentriolar material assembly and the timing of neurogenesis. *Neuron* 65, 191-203, doi:10.1016/j.neuron.2010.01.011 (2010).

This being said, that Pcm1 mutant has very interesting cilia anomalies and most probably neuroanatomical defects that authors failed to appreciate. This raises several interesting cell biological questions. Previous papers mentioning Pcm1 were mostly carried out in vitro. Therefore, a careful analysis of the mutant phenotype would lead to very significant contributions and progress in understanding the role of that protein in development and in adult. The analysis of the mutant phenotype is so preliminary that the rest of the paper and the hard work in RNA seq, analysis of variants in human families and work in zebrafish, by not being really based on strong phenotypic traits to guide them, conveys the unavoidable impression that authors do not follow their data but tend to have some a priori in terms of human diseases.

Again, we regret to have left that impression. We hope our new mouse data changes the reviewer's opinion. Of course, this is just the beginning of our work on this mouse, and we look forward to sharing this data with the community as we continue to study it even more extensively.

Reviewer #2

In this study, Oh et al. investigate the in vivo functions of PCM1, a gene involved in cilia biogenesis. They discover that the loss of PCM1 leads to numerous behavioral phenotypes in adult mice that are relevant to symptoms of schizophrenia, such as hyperactivity, impaired prepulse inhibition, anhedonia, and impaired spatial learning and memory. Furthermore, they find that progressive ciliary defects develop postnatally in mice lacking PCM1, concurrent with the onset of several behavioral phenotypes. Transcriptomic profiling shows the dysregulation of GPCR and amine pathways, although the significance of these changes with respect to disease is unclear. Through the sequencing of Danish cohorts, the authors also link schizophrenia cases with ultra-rare variants of PCM1 and have tested their pathogenicity in an in vivo complementation assay in zebrafish. Overall, this is a potentially important study, revealing an interesting role for PCM1 in mouse neural development and an association between rare PCM1 variants and schizophrenia, although the disease-causing mechanisms involving this gene remain largely unknown. The authors draw numerous conclusions based on loosely-connected evidence, such as alteration of D2R and catalepsy difference upon Haloperidol treatment, and multiple technical details are missing, making it difficult to evaluate the validity of the presented data.

We appreciate the reviewer's comments. We have addressed the criticism with new considerable new data and clarification of the points relating our data to the human disease.

1. The authors state that "Pcm1^{-/-} mice were hyper-responsive to PCP." However, Figure 2C does not show that Pcm1^{-/-} mice demonstrate an over-exuberant response in comparison to wild type. To draw this conclusion, the authors will need to show comparisons of vehicle- and PCP-injected wild type and Pcm1^{-/-} mice at various doses.

We apologize for the lack of clarity. We now include quantification of the fold change above vehicle-treated in the text, and we added the vehicle-treated animal responses as part of Figure 2c.

Furthermore, the authors did not discuss why they chose PCP as opposed to the better-tolerated (and more commonly used) ketamine, which provides a more specific NMDAR antagonism. Finally, would the hyperactivity phenotype be a significant confound to their PPI measurement?

We share the reviewer's concern - indeed, we were aware of potential confounding hyperactivity. We include a comment on this point in the text, "mindful of the possible confounding effect of increased activity in PCM1 KO mice, we measured PPI null activity and found no detectable difference between genotypes," with data shown in Figure S3a.

2. The authors do not mention that PCP can also act as a 5-HT reuptake inhibitor, and that haloperidol and clozapine are also antagonists of certain subunits of the 5-HT receptor thereby potentially affecting the negative and cognitive symptoms associated with SZ by altering serotonin signaling. The link to amine pathway signaling in their RNAseq data further encourages the examination of this

receptor group within Pcm1^{-/-} mice and their role in centriolar protein-associated SZ cases. It would be interesting to assess some of these phenotypes using a higher dose of haloperidol (>2.6mg/kg) to antagonize the 5-HT receptor to significant levels.

Upon further consideration, we agree with the reviewer. Due to the imperfection of the assay, this experiment adds nothing to the paper and brings up more questions than it resolves. Therefore, we have removed it from the manuscript.

3. The authors state that “Subsequent to validation of the RNAseq-derived misregulation of D1R and D2R by qRT-PCR in Pcm1^{-/-} mice...” However, there is no reference to this data in the paper and it does not appear to be included. The authors also state in multiple places, including the abstract, about aberrant localization of D2Rs. Where is the data that show mis-localization of D2Rs?

We apologize for the mix-up. We intended to reference the protein level validation of the RNA-seq with the radioligand assay, which is an important functional validation of altered gene expression. The text has been modified to reflect the data.

In a previous version of the manuscript, we showed D2R localization. However, we were not ultimately satisfied with the quality of that imaging data, so we decided to remove it, and we failed to delete the references to it from the text; we are embarrassed by this error. The revised manuscript has been corrected.

4. Suppl. Fig. 1D is of poor quality. How can a conclusion be drawn from this figure? How do the authors know the staining is specific and meaningful?

We agree, and we apologize for including the low-resolution version. We have replaced the images with higher quality images with arrows, and we have included PCM1 staining from mouse brain tissue to further illustrate the point (Figure S1e). We point out that PCM1 sub-cellular localization is diffuse and punctate^{1,2}, so without extraordinary imaging capabilities, improving the resolution is difficult. We are also confident in our staining due to the fact that the pattern we see is absent in the PCM1 KO samples.

1. **Figures 3f and 6i.** Holdgaard, S. G. et al. Selective autophagy maintains centrosome integrity and accurate mitosis by turnover of centriolar satellites. *Nat Commun* 10, 4176, doi:10.1038/s41467-019-12094-9 (2019).
2. **Figures EV1, 2c, and 3c.** Gheiratmand, L. et al. Spatial and proteomic profiling reveals centrosome-independent features of centriolar satellites. *EMBO J* 38, e101109, doi:10.15252/embj.2018101109 (2019).

5. Figure 1B presents a rather important finding, but additional measurements, such as brain size, brain weight, thickness of cortical layers, need to be carefully measured and presented.

We appreciate the suggestions. Indeed, the mouse anatomical phenotype could have been quantified more completely. Therefore, we performed extensive new analysis, leveraging our existing MRI data and new histology. These new data are presented in Figure 1 and Figure S2.

6. Figure 3 also represents a key finding, but how are bulbous cilia measured and quantified? Representative images at P90-120 should be included as well. Similarly, how is the data for Suppl. Fig. 7 defined and quantified?

Abnormal or “bulbous” cilia are a hallmark cellular phenotype in several ciliopathies. Cilia are stained and then classified as “normal” or not based on the morphology. Cilia with base width greater than twice that of the distal cilia are considered abnormal, as are cilia with no clear extension beyond the base. We now included more images of representative cilia from adult mice (Figure. S5b), and we more explicitly reference the existing adult cilia imaging (Figure S5c and Figure S10).

Our remark that SSTR3 and MCHR1 localized to the cilia was based on the consistent overlapping fluorescence intensity of the immunostained proteins. Representative intensity profile quantile-quantile plots have now been included with Pearson correlation statistics. All of these data are now presented in Figure S10.

7. Progressive appearance of ciliary defects and progressive development of behavioral phenotypes are two key findings revealed by this study. But, the progressive nature of behavioral phenotypes is not well conducted. Other than PPI, locomotion, anhedonia, and learning and memory tests should be included for juvenile or prepubescent mice to corroborate the conclusion.

We fully agree with this suggestion. We were able to perform these additional behavioral analyses on developing mice. In P21 mice, we now show novel object recognition tests, in addition to PPI and locomotion. We also include new P40 anhedonia and novel object recognition analysis. This is all shown in the new, Figure S6. In future work, we look forward to exploring how the sequence of cell biological and neuroanatomical changes manifest as behavioral differences in these animals.

8. Is there any *in vivo* evidence supporting the association of D2R with PCM1?

We appreciate the reviewer's question and understand the limitations of our previous cell culture-based approach. We have now included an IP western blot from mouse brain homogenates that show an *in vivo* association between Pcm1 and the D2 receptor, in addition to the prior work. These data are now included in the revised Figure 4c.

9. How was catalepsy measured and quantified?

We apologize for the lack of clarity. All the behavioral testing was previously documented in one awkward block in the methods section. We have broken that section up to improve readability. Prepulse inhibition, novel object recognition memory testing, morris water maze, anhedonia, and catalepsy are now separate sections in the methods. As stated in the manuscript, catalepsy was assessed using the horizontal bar test. The forepaws of the mice were placed on the bar and the latency to remove the paws was scored. Subsequently, mice were injected with different doses of HP and re-tested for catalepsy 1 hr after injection. The data are presented as time spent in catalepsy with a 60-sec cut-off.

Reviewer #3

The manuscript provides evidence for the involvement of genetic variation in the pericentriolar material 1 (PCM1) gene in susceptibility to schizophrenia. The gene has previously been reported to be associated with the disease but these findings did not show what would now be considered to be strong evidence for involvement in schizophrenia. There was also prior evidence that reduced expression of PCM1 led to impaired cortical migration.

The manuscript draws on evidence from sequence analysis of 880 severely affected schizophrenia subjects and 863 control subjects from Denmark. This sequencing identified a number of rare and "ultra-rare" variants. In aggregate these variants showed modest evidence for association with schizophrenia. However, by modelling the ability of the mutations to recover a phenotype in zebra fish embryos the authors were able to discriminate between those genetic variants that were considered to be benign and those that were considered to be "pathogenic". Using this categorisation of the variants and selecting only the pathogenic variants the authors were then able to show evidence for association with schizophrenia risk ($p=4.5 \times 10^{-5}$, OR=44.6). The authors also present compelling and interesting data that a PCM1 mutant mouse shows brain structural changes similar to those reported in a PCM1 association study in humans. They also present detailed pharmacological, behavioural and gene expression data that suggest that ablation of PCM1 has profound impacts on processes thought to be linked to schizophrenia.

I believe that the manuscript provides important new evidence implicating the PCM1 gene in

schizophrenia risk. The multiple lines of high quality data do not in isolation, present definitive evidence that the gene is involved in schizophrenia. However, the evidence from the manuscript combined with the prior findings provides good evidence that this gene may be involved in schizophrenia risk.

We thank the reviewer for the positive comments on our work and for carefully reading our manuscript.

The authors should however provide more details on the human genetic variants that they have identified. They do not provide the genomic locations of the variants nor the nucleotide changes. It would also be useful to know whether these variants have been reported before in GnomAD (<http://gnomad.broadinstitute.org/>) or in the ExAC samples and in the non-psych ExAC samples (<http://exac.broadinstitute.org/>).

We thank the reviewer for the suggestion, and agree it improves the presentation of the data. The genomic locus, nucleotide position, and the allele frequencies of those annotated in gnomAD have now been included in Supplemental Table 1.

As stated above, the reported evidence for schizophrenia risk from combined analysis of the pathogenic variants is ($p=4.5 \times 10^{-5}$, OR=44.6). The authors should comment as to whether this finding exceeds a genome wide threshold of significance and/or indeed what the false positive rate would be in context of this form of analysis.

We hesitate to apply a genome-wide significance threshold to this type of analysis, since formally determining it would require repeating the experiment on every human gene. However, since this is a gene-based test, we could reasonably approximate the genome-wide significant threshold to be $p \approx 1.7 \times 10^{-6}$, or $0.05/30,000$ (roughly the number of human genes).

REVIEWERS' COMMENTS:

Reviewer #1 (Remarks to the Author):

Authors have generated additional data and tried to answer the criticism I made about the previous version. Even though those new data and analysis does not answer all my questions, the paper is clearly improved. It is an elegant contribution.

Reviewer #2 (Remarks to the Author):

In this revised manuscript, the authors have conducted numerous additional experiments, re-organized data presentation, and provided collective evidence reporting that mice lacking Pcm1 show neuroanatomical alteration, behavioral deficits, and abnormalities in cilia morphology. They also find amine and GPCR pathway related genes are altered in three different brain regions of Pcm1 null mice, and PCM1 biochemically associates with D2R. Finally, through sequencing studies of two Danish cohorts and complementation studies in zebrafish, they report an association of rare variants in PCM1 with severe form of schizophrenia. Overall, the quality of data presented in this revision is much more improved compared to the initial submission, and these data support each of their conclusions in general, but lacking connections from one to another. In my view, the study remains descriptive in nature and falls short in providing sufficient mechanistic insights linking dysfunction of PCM1 to many of the behavioral, anatomical and even cilia phenotypes they showed in this manuscript. The connection to schizophrenia, other than genetics, is far apart. I have a hard time to justify the acceptance of this study to Nature Communications.

Response to Reviewer Comments

Reviewer #1 (Remarks to the Author):

Authors have generated additional data and tried to answer the criticism I made about the previous version. Even though those new data and analysis does not answer all my questions, the paper is clearly improved. It is an elegant contribution.

We thank the reviewer for considering our manuscript and for the supportive comments.

Reviewer #2 (Remarks to the Author):

In this revised manuscript, the authors have conducted numerous additional experiments, re-organized data presentation, and provided collective evidence reporting that mice lacking Pcm1 show neuroanatomical alteration, behavioral deficits, and abnormalities in cilia morphology. They also find amine and GPCR pathway related genes are altered in three different brain regions of Pcm1 null mice, and PCM1 biochemically associates with D2R. Finally, through sequencing studies of two Danish cohorts and complementation studies in zebrafish, they report an association of rare variants in PCM1 with severe form of schizophrenia. Overall, the quality of data presented in this revision is much more improved compared to the initial submission, and these data support each of their conclusions in general, but lacking connections from one to another. In my view, the study remains descriptive in nature and falls short in providing sufficient mechanistic insights linking dysfunction of PCM1 to many of the behavioral, anatomical and even cilia phenotypes they showed in this manuscript. The connection to schizophrenia, other than genetics, is far apart. I have a hard time to justify the acceptance of this study to Nature Communications.

We thank the reviewer for considering our manuscript. While we disagree with the sentiment that the manuscript is “descriptive in nature” and with that general ethos, we considered the point and have been mindful to minimize any perceived over-interpretation of the data.